# HelpSteer2: Open-source dataset for training top-performing reward models

**Zhilin Wang, Yi Dong, Olivier Delalleau, Jiaqi Zeng, Gerald Shen**
**Daniel Egert, Jimmy J. Zhang, Makesh Narsimhan Sreedhar, Oleksii Kuchaiev**
NVIDIA
`{zhilinw, yidong}@nvidia.com`

## Abstract

High-quality preference datasets are essential for training reward models that can effectively guide large language models (LLMs) in generating high-quality responses aligned with human preferences. As LLMs become stronger and better aligned, permissively licensed preference datasets, such as Open Assistant, HH-RLHF, and HelpSteer need to be updated to remain effective for reward modeling. Methods that distil preference data from proprietary LLMs such as GPT-4 have restrictions on commercial usage imposed by model providers. To improve upon both generated responses and attribute labeling quality, we release HelpSteer2, a permissively licensed preference dataset (CC-BY-4.0). Using a powerful Nemotron-4-340B base model trained on HelpSteer2, we are able to achieve the SOTA score (92.0%) on Reward-Bench's primary dataset, outperforming currently listed open and proprietary models, as of June 12th, 2024. Notably, HelpSteer2 consists of only ten thousand response pairs, an order of magnitude fewer than existing preference datasets (e.g., HH-RLHF), which makes it highly efficient for training reward models. Our extensive experiments demonstrate that reward models trained with HelpSteer2 are effective in aligning LLMs. Additionally, we propose SteerLM 2.0, a model alignment approach that can effectively make use of the rich multi-attribute score predicted by our reward models. HelpSteer2 is available at `https://huggingface.co/datasets/nvidia/HelpSteer2` and code is available at `https://github.com/NVIDIA/NeMo-Aligner`.

## 1 Introduction

Since the pioneering works on Reinforcement Learning from Human Feedback [1, 2], the significance of incorporating preference information into model alignment has been consistently demonstrated [3, 4]. Both proprietary models (e.g., GPT-4 [5], Claude [6], Gemini [7]) and open-source models (e.g., Llama 3 [8], Mistral [9], and Yi [10]) have benefited from preference modeling techniques. However, most of these models lack detailed information about the preference data used in their training, hindering the broader community from fully leveraging these techniques. For instance, Llama 2 [4] only disclosed the use of over 1 million binary comparisons, while Llama 3 [8] reported the use of 10 million data samples for supervised fine-tuning and preference modeling without additional details.

To address this issue, a few domain-general chat preference datasets have been made available to the community. Some of these datasets come with permissive licenses, such as Anthropic's Helpful-Harmless RLHF (MIT license) [2], Open Assistant (Apache 2.0) [11], and HelpSteer (CC-BY-4.0) [12], facilitating their use in both academic and commercial settings. However, these datasets have become less relevant for training the most well-aligned models currently available [13, 14].

38th Conference on Neural Information Processing Systems (NeurIPS 2024) Track on Datasets and Benchmarks.

Others in the community have tackled this challenge by using proprietary models, such as GPT-4, to create preference datasets like Ultrafeedback [15], Nectar [16], and Distilabel-Capybara [17]. Although these datasets are more effective for aligning models, their use is often restricted to academic or non-commercial settings. The terms of use from various large language model providers explicitly prohibit the use of model outputs to develop competing models, posing legal risks for commercial organizations that use these datasets to train their own large language models. Further discussion on existing preference datasets is provided in Appendix 5.

We propose HelpSteer2, a CC-BY-4.0-licensed open-source helpfulness dataset, designed to train state-of-the-art reward models. Additionally, we provide detailed information about our data collection process to aid similar efforts and demonstrate how reward models trained with HelpSteer2 can align large language models with human preferences. We extend SteerLM [18, 12] to present SteerLM 2.0, a novel model alignment paradigm that can effectively utilize multi-faceted rewards from our reward models to train models to follow complex multi-requirement instructions. By open-sourcing the dataset with minimal usage restrictions, we invite the community to utilize and build upon it to develop well-aligned AI systems.

## 2 Dataset

### 2.1 Dataset Collection

**Prompt Collection**   Most of the prompts (over 95%) used in HelpSteer2 are sourced from ShareGPT [19], a platform where ChatGPT users voluntarily share their conversations. We selected this dataset as a source of prompts because we believe that ShareGPT encompasses a diverse range of real-world LLM use cases. Importantly, we only use user inputs from this dataset, while Assistant turns are stripped out to avoid potential model-specific licensing restrictions. We supplemented ShareGPT prompts with a small proportion of proprietary prompts, primarily focused on use cases such as summarization, closed question answering, and extraction. These use cases are relevant in enterprise settings but are less likely to be represented in ShareGPT. An analysis of the distribution of topics represented by these prompts are in Appendix D.

Given that our annotator pool consisted solely of US-based annotators fluent in English and not expected to be competent in other languages, we removed all non-English prompts, as identified by FastText [20]. Additionally, since our annotators lacked expertise in the coding domain, we employed simple heuristics to filter out prompts containing snippets of popular programming languages.

To ensure a diverse sample of prompts, we utilized BERTopic [21] to cluster similar prompts into approximately 1000 topics. We then sampled uniformly from each topic across various deliveries to our vendor. Additionally, we observed that high-quality generation in real-world settings requires the model to handle complex prompts, sometimes containing multiple requirements. Inspired by [22], we assessed the complexity of each prompt on a Likert-5 scale using Nemotron-2-43B [12], with details provided in Appendix E. Subsequently, we sampled prompts uniformly across each complexity level, except for the highest complexity level, which was given twice the weight of other levels.

**Multi-turn Prompt Completion**   To ensure HelpSteer2 is effective for predicting rewards in multi-turn conversations, we included multi-turn prompts, which comprise approximately 29% of the samples. For these prompts, we did not use the original ShareGPT Assistant responses, as those may be generated by models with restrictive licenses. Instead, we replaced these Assistant turns with responses generated by a 22B in-house model, specifically trained to provide Assistant responses given only user turns. This model was fine-tuned using conversations from the Open Assistant [11] and HH-RLHF [2] datasets (see Appendix F for details).

**Response Generation**   We generate two responses per prompt, instead of four as in HelpSteer [12], to minimize annotators' cognitive load during annotation, thereby enhancing rating quality. The sources and associated proportions of these responses are as follows (the two responses for each prompt always come from two different sources):

1. Our own internal LLMs from three generations of models:
   - Nemotron-2 (43B models based on the model used to generate HelpSteer responses, described in more details in [12]): 18.9% of the responses

- Nemotron-3 (8B and 22B models – see [23] for information on the publicly released 8B models. 22B models come from a base model following a similar architecture and pre-training scheme but larger size: 40.4% (2.2% from 8B, 38.2% from 22B)
- Nemotron-4 (15B and 340B models – see [24] for details on the 15B pre-trained model, while the 340B one follows a similar architecture and pre-training scheme but with more parameters): 26.9% (9.5% from 15B, 17.4% from 340B)

2. Mixtral-8x7B-Instruct-v0.1 [9]: 7.9%
3. Human annotators from Scale AI: 5.9%

Throughout the data collection effort, we used aligned versions of the internal models mentioned above, all trained on datasets with permissive commercial licenses using Megatron-LM [25] for pre-training and NeMo-Aligner [26] for fine-tuning. For fine-tuning, we employed several techniques: Supervised Fine-Tuning, SteerLM [18], Reinforcement Learning from Human Feedback [1, 27], and Direct Preference Optimization [28]. This diversity in model sizes and learning algorithms was intended to substantially increase response diversity compared to the original HelpSteer dataset [12], which relied on a single Nemotron-2 43B model for responses. Additionally, we leveraged SteerLM's controllable generation capabilities to generate some responses with randomly sampled SteerLM labels, further varying response styles.

**Response Annotation**    Our response annotation process, guidelines and annotator screening are primarily derived from HelpSteer guidelines [12]. Specifically, for each response, we annotate five attributes (helpfulness, correctness, coherence, complexity, and verbosity) on a Likert-5 scale. However, we have implemented several improvements to the annotation process.

First, we required at least three annotators to annotate each response compared to only one annotator in HelpSteer [12]. We opted for multiple annotators per sample because initial explorations indicated that annotation quality, measured by inter-annotator agreement, is crucial for model training. Without high-quality annotations, the data can be noisy, which can potentially confuse the model on what characterizes a higher score. Each sample is initially annotated by three annotators. If these annotators demonstrate a high level of disagreement (i.e. the difference in helpfulness among them is greater than 2), two additional annotators are recruited to annotate the sample. Overall, samples were on average annotated by 3.41 annotators.

In addition, annotators were asked to rate two responses to the same prompt sequentially. Our initial analysis showed that doing this can allow the annotator to provide a more calibrated score for each response (e.g. if response A is much better than response B, then helpfulness for the response should be much higher). It does so by reducing the likelihood of annotators doing slipshod annotations and also facilitates quality assurance on such annotations. Overall, this means that each sample annotated for HelpSteer2 required substantially more effort and resources compared to HelpSteer [12]. To meet this challenge, we engaged approximately 1,000 US-based annotators through our vendor Scale AI compared to 200 annotators engaged in HelpSteer [12]. We would like to highlight that our guidelines explicitly ask annotators to skip a sample if it contains any Personally Identifiable Information (e.g. name, address, SSN, email, phone numbers) and to flag it for unsafe content (e.g. harmful content, illegal activities, profanity, bias and stereotyping). Please refer to Appendix C for ethical considerations relating to such annotations and Appendix G for the full annotation guidelines.

| *Attribute* | Initial Collection | After Improvements | Post-Processing |
|---|---|---|---|
| Helpfulness | 0.465 | 0.706 | 0.791 |
| Correctness | 0.472 | 0.715 | 0.793 |
| Coherence | 0.169 | 0.387 | 0.428 |
| Complexity | 0.293 | 0.416 | 0.427 |
| Verbosity | 0.342 | 0.536 | 0.548 |

Table 1: Inter-annotator Agreement (quadratic weighted Cohen's $\kappa$) for HelpSteer2 attributes.

We measure inter-annotator agreement using Quadratic weighted Cohen's $\kappa$ [29]. Compared to metrics for measuring more than two annotators (e.g. Krippendorff's $\alpha$ or Fleiss' $\kappa$), we chose to use Cohen's $\kappa$ because given the large number of annotators (1000), multiple individual annotators were rarely allocated common sets of samples to annotate. We also chose to use the quadratic

weighted version of Cohen's $\kappa$ [30] because HelpSteer2 attributes are ordinal scores, meaning that disagreements between 0 and 4 should be penalized much more heavily compared to between 0 and 1. Initial annotations tend to have low inter-annotator agreement (e.g. Cohen's $\kappa = 0.465$ for helpfulness) as seen in Table 1. Throughout the annotation process, we made several improvements with our vendor, clarifying how our guidelines apply to various edge cases (e.g., whether coherence should consider previous turns, and how helpfulness should be evaluated if prompt instructions are unclear). We used up to five annotators per sample but only retained annotations from the three most in agreement. After the annotations, our vendor performed extensive quality assurance, with each annotation undergoing a minimum of two human reviews in addition to automated checks. Part of the quality assurance process involved removing annotations from annotators who were deemed 'untrusted' or consistently had low agreement with others. These efforts improved inter-annotator agreement for all attributes, with Cohen's $\kappa$ for helpfulness reaching 0.706.

As a final step, we retained only responses for which the differences in helpfulness attribute among annotators were 2 points or below on a Likert-5 scale (for both responses to a common prompt), resulting in the removal of about 10% of the samples. The 2-point threshold was chosen to balance the proportion of retained data and the relative noise in these annotations, recognizing that differences among annotators can also stem from inherent subjectivity or individual preferences rather than misunderstandings of the annotation task. Extensive filtering of annotations was performed by both our vendor and the research team at various stages, with approximately 50% of all annotations ultimately excluded from the dataset. Our final dataset contains 21,362 high-quality annotated samples, consisting of 10,681 prompts each with two annotated responses. The dataset is divided into a training subset (95% of the data) and a validation subset (5% of the data).

## 2.2 Dataset Analysis

As shown in Table 2, model responses in HelpSteer2 are more helpful, correct, coherent, verbose, and complex due to stronger models used for response generation. The most substantial change is on the coherence attribute, reaching 3.63 out of a full score of 4 on a Likert-5 scale, meaning that generating coherent responses is no longer a challenge for the stronger models. In addition, the verbosity attribute also increased by almost 0.5 from 1.53 to 2.00, meaning that responses changed from being terse to having a good spread of concise and verbose responses. The increase in average response length by 3x from 497.3 to 1492.6 characters also supports this observation.

| Attribute | Mean | | Standard Deviation | | Pearson's R with Helpfulness | |
|---|---|---|---|---|---|---|
| | HS | HS2 | HS | HS2 | HS | HS2 |
| Helpfulness | 2.7856 | 2.8655 | 0.9793 | 1.2703 | 1 | 1 |
| Correctness | 2.8369 | 2.9644 | 0.9935 | 1.2689 | 0.8525 | 0.9430 |
| Coherence | 3.2991 | 3.6393 | 0.7699 | 0.6491 | 0.6348 | 0.4979 |
| Complexity | 1.4423 | 1.7048 | 0.8205 | 0.6986 | 0.2361 | 0.1805 |
| Verbosity | 1.5331 | 1.9999 | 0.9287 | 0.7571 | 0.2555 | 0.0600 |
| No. of turns in prompt | 1 | 2.8348 | 0 | 3.8221 | - | -0.0520 |
| No. of chars in prompt | 2491.8 | 712.6 | 1701.7 | 877.8 | 0.0337 | -0.0774 |
| No. of chars in response | 497.3 | 1492.6 | 426.7 | 1065.7 | 0.1951 | 0.0845 |

Table 2: Descriptive statistics for attributes in HelpSteer (HS) and HelpSteer2 (HS2). Please refer to [12] for comparison with Open Assistant and HH-RLHF. Scores for each attribute are between 0 and 4 on a Likert-5 scale.

On the other hand, although HelpSteer2 contains multi-turn prompts with a mean of 2.83 turns compared to only single-turn prompts in HelpSteer, the average character length of prompts in HelpSteer2 is 712 characters, a fraction of the 2491 characters in HelpSteer. This difference is likely because HelpSteer2 prompts are more conversational and succinct, primarily based on ShareGPT, whereas HelpSteer prompts are exclusively based on enterprise use cases involving context documents such as summarization, closed question answering, and extraction.

In Table 2, we observe that coherence is a much weaker predictor of helpfulness in HelpSteer2 (Pearson's R=0.4979) compared to HelpSteer (Pearson's R=0.6348). This is likely due to the distribution of coherence scores, as most responses in HelpSteer2 are coherent given the use of stronger models. Conversely, correctness has become a stronger predictor of helpfulness in HelpSteer2

(Pearson's R=0.9430) than in HelpSteer (Pearson's R=0.8525). This likely occurs because, with all responses being highly coherent, factuality becomes a more critical factor in determining overall helpfulness. Additionally, the Pearson's R values for both complexity (0.2361 to 0.1805) and verbosity (0.2555 to 0.0600) have decreased, indicating that annotators are less influenced by the complexity and verbosity of responses when assessing overall helpfulness in HelpSteer2. This is beneficial for reward model training, as models can learn that generating complex and verbose responses does not substantially contribute to being helpful.

Helpfulness is also slightly negatively correlated with prompt character length (Pearson's R=-0.0774) and prompt turns (Pearson's R=-0.0520). This suggests that models used for response generation are likely to perform worse in generating follow-up responses compared to initial responses, a trend observed in many models in MT Bench [31]. Finally, response length is slightly positively correlated with helpfulness (Pearson's R=0.0845), consistent with the correlation between verbosity and helpfulness (Pearson's R=0.0600).

## 3 Reward Model

**Training** We follow the training approach of SteerLM Regression Reward Models [12], the format for which HelpSteer2 data was collected in. SteerLM Regression Reward Modeling aims to predict the values of various attributes as annotated by humans (e.g. helpfulness=2; correctness=3). There are alternative reward modeling methods such as Bradley-Terry [1, 2, 27], but exploring such methods are left to future work as HelpSteer2 data was *specifically collected* for SteerLM Regression Reward Modeling.

We train reward models consisting of a base model and a linear layer that converts the final layer representation of the end-of-response token into five scalar values, each corresponding to a HelpSteer2 attribute. The reward models are trained on top of two open-source base models: Llama 3 70B and Nemotron-4 340B[1] (described in Sec. 2.1). For each model, we train for two epochs using HelpSteer2 data, with a global batch size of 128. We select the top checkpoints with the lowest validation loss for evaluation. We train with a MSE loss function, a constant learning rate on each model (70B: 2e-6, 340B: 7e-7) using an AdamW optimizer [32] and 10 warmup steps, following a LR search (70B: {1,2,3,4,5}e-6; 340B: {1,3,5,7,9}e-7). For comparison, we also trained a Llama 3 70B base model separately using 1 epoch of HH-RLHF [2]; 1 epoch of Open Assistant [11] or 2 epochs of HelpSteer [12] (to approximately match for difference in dataset size) using the same hyper-parameters.

**Evaluation** Following [33, 34], we evaluate the trained reward models using Reward Bench [14] excluding the optional Prior Sets category which we report separately (with detailed reasons in Appendix H). Reward Bench comprises 2985 diverse tasks, each consisting of a prompt, a chosen response, and a rejected response. Task accuracy is calculated based on whether the chosen response receives a higher reward than the rejected response. The tasks in Reward Bench are categorized into four main categories: Chat, Chat-Hard, Safety, and Reasoning. Overall accuracy is determined by taking the mean of each category. Details for evaluation are in Appendix H. We choose to use RewardBench due to its diversity of tasks (4 categories and 23 sub-categories), which minimizes the likelihood of overfitting. With over 80 models on the leaderboard [35] available for comparison, it serves as a well-trusted benchmark.

**Results** Overall, reward models trained with HelpSteer2 perform well on Reward Bench, achieving state-of-the-art numbers compared to proprietary models and those trained with data allowing permissive use. This is particularly noteworthy given that HelpSteer2 consists of only 10k response pairs. Llama 3 70B trained on HelpSteer2 (88.8% Overall)[2] outperforms all other models trained with data allowing permissive use by >9.7%, including the same Llama 3 70B base model trained with Open Assistant, HH-RLHF or HelpSteer. Scaling up the base model to Nemotron-4 340B with the same dataset results in the trained reward model topping the Reward Bench primary leaderboard with an overall performance of 92.0%[3]. This suggests that as more capable base models emerge, training them with HelpSteer2 can lead to more powerful reward models. We measure RewardBench using the weighted sum of five attribute values with the weights [0.3, 0.74, 0.46, 0.47, -0.33]

---

[1]Available at `https://huggingface.co/nvidia/Nemotron-4-340B-Base`
[2]Available at `https://huggingface.co/nvidia/Llama3-70B-SteerLM-RM`
[3]Available at `https://huggingface.co/nvidia/Nemotron-4-340B-Reward`

| Source of Model/Training Data | Model | Reward Bench Primary Dataset | | | | | Prior Sets |
|---|---|---|---|---|---|---|---|
| | | Overall | Chat | Chat-Hard | Safety | Reasoning | |
| *Proprietary Models* | **Nemotron-4-340B-Reward** | **92.0** | 95.8 | **87.1** | 91.5 | 93.7 | 67.4 |
| | Cohere May 2024 | 89.5 | 96.4 | 71.3 | **92.7** | 97.7 | **78.2** |
| | Gemini 1.5 Pro-0514 | 88.1 | 92.3 | 80.6 | 87.5 | 92.0 | - |
| | Cohere March 2024 | 87.1 | 94.7 | 65.1 | 90.3 | **98.2** | 74.6 |
| | GPT-4-0125-preview | 85.9 | 95.3 | 74.3 | 87.2 | 86.9 | 70.9 |
| | GPT-4-0409-preview | 85.1 | 95.3 | 75.4 | 87.1 | 82.7 | 73.6 |
| | GPT-4o-0513 | 84.7 | **96.6** | 70.4 | 86.7 | 84.9 | 72.6 |
| | Claude-3-Opus-02292024 | 80.7 | 94.7 | 60.3 | 89.1 | 78.7 | - |
| *Trained with GPT-4 Generated Data* | ArmoRM-Llama 3 8B | **90.8** | 96.9 | **76.8** | 92.2 | 97.3 | 74.3 |
| | RLHFlow-Llama 3 8B | 87.1 | **98.3** | 65.8 | 89.7 | 94.7 | **74.6** |
| | Eurus RM Mistral 7B | 82.8 | 98.0 | 65.6 | 81.2 | 86.3 | 71.7 |
| | Starling RM Yi 34B | 82.7 | 96.9 | 57.2 | 88.2 | 88.5 | 71.4 |
| | Prometheus 2 Mistral 8x7B | 75.3 | 93.0 | 47.1 | 83.5 | 77.4 | - |
| *Trained with Data allowing Permissive Use* | **Llama 3 70B RM** (w. HelpSteer2)* | **88.8** | 91.3 | **80.3** | **92.8** | **90.7** | 66.5 |
| | Llama 3 70B (w. Open Assistant)* | 79.1 | 91.3 | 59.2 | 76.0 | 89.9 | 66.7 |
| | Llama 3 70B Instruct | 76.0 | **97.6** | 58.9 | 69.2 | 78.5 | **70.4** |
| | Llama 3 70B (w. HH-RLHF)* | 73.9 | 94.4 | 54.6 | 81.2 | 65.6 | 68.8 |
| | Pythia 1.4B (w. Open Assistant) | 70.0 | 88.5 | 48.7 | 65.3 | 77.5 | 65.3 |
| | Llama 3 70B (w. HelpSteer)* | 66.1 | 93.3 | 59.7 | 56.8 | 54.9 | 67.7 |

Table 3: Performance of Models on Reward Bench. Higher is better for each category. All numbers except models trained by us (which are marked with *) are taken from Reward Bench leaderboard.

for 340B model and [0.65, 0.8, 0.45, 0.55, -0.4] for 70B model, based on a search for optimal weights at 0.01 granularity. For the 340B model, searching at 0.1 granularity reduces overall Rewardbench by 0.1% while 1 granularity reduces it by 0.2% and using the helpfulness attribute only reduces it by 1.0%.

Relative to other models, those trained with HelpSteer2 perform exceedingly well in the Chat-Hard category, surpassing the second-best by 6.5%. This is because HelpSteer2 is primarily aligned with the task of distinguishing between good and excellent responses. Chat-Hard is likely the most relevant metric for preference learning with capable domain-general LLMs since we typically start with a good model and aim to improve its responses further. Unexpectedly, models trained with HelpSteer2 also show good performance in the Safety and Reasoning categories, even though HelpSteer2 does not explicitly focus on these aspects. This may be due to an implicit association between helpful responses and general safety, and transfer learning between being factually correct and reasoning tasks. However, HelpSteer2 trained models do not surpass the Reasoning performance of the strongest alternative models, which are trained on specific reasoning datasets, such as UltraInteract [34]. Finally, HelpSteer2 trained models substantially under-perform many other models on Prior Sets, likely because those other models were trained on the training subsets of these Prior Sets [33].

# 4   Aligned Models

We demonstrate three approaches for using the Llama 3 70B Reward Model to align LLMs: Iterative Direct Preference Optimization (Iterative DPO), Proximal Policy Optimization (PPO) and SteerLM.

## 4.1   Evaluation

Following HelpSteer [12], we use MT Bench [31] to measure helpfulness, TruthfulQA MC2 [36] to measure correctness, and the mean number of characters in MT Bench responses to measure verbosity. However, instead of the GPT-4-0613 judge used in HelpSteer [12], we use GPT-4-0125-Preview (Turbo) as a judge because we find that it is a stronger model and better suited as a judge. In addition, we also use AlpacaEval 2.0 Length Controlled [37] and Arena Hard [38] as secondary measures of helpfulness, following [33, 39]. MT Bench is also referenced as a validation metric for checkpoint selection. Details for each evaluation metric is available in Appendix H.

## 4.2   SFT

Supervised Fine-tuning (SFT) trains a model through demonstrations, serving as a prerequisite for preference learning methods such as DPO in Sec. 4.3 and PPO in Sec. 4.4. Following HelpSteer [12], we train a Llama 3 70B Base model using only Open Assistant [11] with 56k conversations for 2400 steps with a global batch size of 128 (close to 4 epochs). We use a constant learning rate (LR) of 2e-6 using the AdamW optimizer after searching LR in {1,2,3,4,5}e-6, saving a checkpoint every 200

steps. This represents the SFT model trained on existing open-sourced data only. However, we find that a SFT model trained with only Open Assistant is weak compared to the Llama 3 70B Instruct, likely due to the inconsistent quality of the responses it contains.

Therefore, we trained another model using an SFT dataset (named 'Daring Anteater')[4] consisting of 100k conversations, each averaging 2.88 model turns. Approximately 93% of the data are synthetically generated following a similar pipeline as [40] by replacing OpenAI models with an earlier aligned version of Nemotron-4 340B[5] and Mixtral-8x7B-Instruct-v0.1 [9], while the rest comes from ARB [42], SciBench [43], tigerbot-leetcode [44], PRM800K [45], FinQA [46], and wikitablequestions [47]. We trained this model using identical hyper-parameters except training it for 1600 steps with a global batch size of 384 (close to 2 epochs), given the larger size of the dataset. All models on DPO and PPO are trained starting from this model.

## 4.3   DPO

Direct Preference Optimization [28] is a simple and stable approach to performing preference-based model alignment without requiring the complex setup of Reinforcement Learning from Human Feedback (RLHF). We first performed DPO training on the SFT model from Sec. 4.2. To do this training, we converted our HelpSteer2 train set into a preference dataset by taking the response with the higher helpfulness score as the chosen response, with the remaining response being the rejected response. In cases where the helpfulness scores were identical, we discarded that pair entirely. This became our HelpSteer2 DPO dataset, which contains 7,221 training samples. We then performed DPO training on this data for 7 epochs using a constant LR of 2e-7, Kullback–Leibler (KL) penalty of 1e-3, AdamW optimizer, Global Batch Size 128, and Weight Decay 0.1. Optimal LR was identified following a search among {3e-7, 2e-7, 1e-7, 9e-8} and KL penalty following a search among {1e-3, 4e-4}. We evaluated checkpoints once every 25 steps.

We then performed Iterative DPO [33] on this model by utilizing 20k prompts from the Daring Anteater SFT dataset and generating 10 responses per prompt (temperature=0.7, top-p=0.9). These responses were then scored by the Llama 3 70B Reward Model (Sec. 3) and a pairwise preference dataset generated by taking the highest and lowest goodness score for the chosen and rejected, respectively. The goodness score is a scalar based on 0.65*helpfulness + 0.8*correctness + 0.45*coherence, which we find to give best differentiation between chosen and rejected responses in RewardBench prompts. We then performed DPO training on this data for 3 epochs using similar hyper-parameters as above, except KL penalty of 1e-3 and LR of 9e-8, following similar hyper-parameter search.

## 4.4   PPO

Proximal Policy Optimization (PPO) [48] is a complex but effective approach to performing Reinforcement Learning from Human Feedback, which contributed to the success of many frontier models include GPT-4 [5]. Starting with the SFT model we trained in Sec. 4.2, we apply PPO similar to the recipe described in [1]. The rewards are provided by the Llama 3 70B Reward Model (Sec. 3) and we sample the policy with HelpSteer2 prompts. In order to regularize the optimization process, we add a KL divergence penalty in the rewards to prevent the policy from deviating too far away from the SFT model. The PPO value model is initialized from the reward model.

The reward was calculated using goodness score (Sec. 4.3), followed by taking away the mean of the HelpSteer2 responses and dividing it by its standard deviation. We trained PPO using a global batch size of 128, a rollout buffer of 128 and a constant LR of 1e-7 and KL-penalty of 3e-3, after searching LR in {1,2,3,4,5}e-7 and the KL-penalty in {1,2,3,4,5}e-3. We train for 64 steps and evaluate a checkpoint every 4 steps. The generation stage of PPO is optimized using NeMo-Aligner's integration of TensorRT-LLM [26].

---

[4]Available at `https://huggingface.co/datasets/nvidia/Daring-Anteater`

[5]While this might be considered as distilling from a larger model, there is no evidence suggesting that Llama 3 70B Instruct was not trained by distilling the announced-but-unreleased Llama 3 400B+ [41] and hence, we believe this is a fair comparison.

## 4.5 SteerLM

SteerLM [12, 18] aligns language models by steering them towards generating outputs with desired attribute values by conditioning on various attributes during training. We trained the SteerLM model following [12] . Specifically, we used the Llama 3 70B Reward Model to annotate the Daring Anteater SFT dataset (Sec. 4.2), followed by attribute-conditioned supervised fine-tuning of a language model on the annotated dataset to generate responses conditioned on target attribute scores.

However, the original SteerLM method does not explicitly enforce the generated responses to follow the desired attribute distribution conditioned on during training. To address this limitation, we propose SteerLM 2.0, which iteratively trains the model to approximate the optimal SteerLM policy constructed by the reward model. This is achieved using the original SteerLM trained model to generate multiple sampled responses and then using a KL divergence loss between current policy and optimal SteerLM policy to guide the model towards generating a response that is more reflective of the desired attribute values. SteerLM 2.0 can be conducted in iterations (n=2) using the optimized policy after each iteration to sample responses and train an improved policy. In each iteration, we sampled multiple diverse responses (n=10, temperature=0.7, top-p=0.9) from 20,000 different prompts from the Daring Anteater SFT dataset. SteerLM 2.0 is trained for 2 epochs with AdamW optimizer constant LR 1e-7 and global batch size 128.

**Method Details** SteerLM 2.0 trains a model $Q_\theta(y|a,x)$ that can generate responses $y$ conditioned on a prompt $x$ and desired attributes $a$, while approximating the optimal conditional distribution $P(y|a,x)$ derived from the optimal reward model $P(a|x,y)$. $P(a|x,y)$ is the attribute prediction model that can be trained on labeled data. To convert the regression reward model into a probabilistic reward model, we use the Beta distribution function to estimate the probability of different reward output levels. We scale the HelpSteer reward model output $r$ to $[0,1]$ and compute the Beta distribution parameters by setting $\alpha = 24r$ and $\beta = 24 - \alpha$. We choose $\alpha + \beta = 24$ as it matches the ground truth distribution of the training data. The probability $P(a = n)$ is calculated as $P_{\alpha,\beta}(X_{i+1}) - P_{\alpha,\beta}(X_i)$, where $P_{\alpha,\beta}$ is the cumulative Beta probability distribution function, and $X_{i+1}$ and $X_i$ are the normalized bin boundaries of the value $n$. Note, the beta distribution approximation is not an integral part of the SteerLM 2.0 method itself, but rather a tool to convert our regression-based reward model into a probabilistic one. Investigations with 'true' probabilistic reward models is left as future work, given that the main goal of this paper is to demonstrate how the Reward Model trained with HelpSteer2 can be used for model alignment.

We first derive the optimal conditional distribution $P(y|a,x)$ using Bayes' rule:

$$P(y|a,x) = \frac{P(a|x,y)P(y|x)}{P(a,x)} \propto P(a|x,y)P(y|x)$$

Here, $P(y|x)$ is the unconditional response distribution from a separate language model (supervised fine-tuning model using Daring Anteater SFT dataset, see Sec. 4.2). The optimal $P(y|a,x)$ can be constructed by combining $P(y|x)$ and $P(a|x,y)$.

To efficiently approximate $P(y|a,x)$, we train a parametric model $Q_\theta(y|a,x)$ by minimizing the KL divergence:

$$\min_\theta \mathbb{E}_{a,x} D_{KL}(P(y|a,x)||Q_\theta(y|a,x))$$

This KL divergence loss can be written as:

$$-\mathbb{E}_{a,x,y \sim P(a)P(x)P(y|a,x)} \log Q_\theta(y|a,x)$$

To optimize the loss, we estimate its gradient using samples from the original SteerLM model $y_i \sim Q'(y|a,x)$:

$$\nabla_\theta L = -\sum_i (w_i' - b_i') \nabla_\theta \log Q_\theta(y_i|a,x)$$

where the $Q_\theta$ is initialized with original SteerLM model $Q'$ during training.

Where $w_i'$ and $b_i'$ are normalized importance weights. This gradient estimator has reduced variance compared to the naive approach [49]. The resulting SteerLM 2.0 model $Q_\theta(y|a,x)$ can generate responses $y$ conditioned on attributes $a$ by approximately following the optimal $P(y|a,x)$ distribution. Further details and derivation of SteerLM 2.0 can be found in Appendix I.

**Inference Attributes**    In this paper, we focus on to calibrate the model to generate good responses, so we choose to focus on one set of desired attributes for response sampling. Because HelpSteer2 responses are much (around 3x) longer and more complex than in HelpSteer [12], we found that using Complexity 2 and Verbosity 2 as default leads to more better generations than setting them both to 4, as done in HelpSteer [12]. The other three attributes (Helpfulness, Correctness and Coherence are set to 4, as in HelpSteer [12].

## 4.6   Results

| Technique | Model | MT Bench (GPT-4-Turbo) | Mean Response Length (Chars.) | TruthfulQA MC2 | AlpacaEval 2.0 LC (SE) | Arena Hard (95% CI) |
|---|---|---|---|---|---|---|
| *Baseline* | GPT-4-0613* | 8.12 | 1057.1 | 0.5900 | 30.20 (1.07) | 37.9 (-2.8, 2.4) |
| | Llama 3 70B Instruct* | 8.16 | 1683.0 | 0.6181 | **34.40** (1.38) | 41.1 (-2.0, 2.2) |
| *SFT* | SFT w. DA | 7.96 | 1514.4 | 0.6025 | 32.87 (1.40) | 39.6 (-2.3, 2.4) |
| *DPO* | DPO w. HelpSteer2 | 8.04 | 1532.1 | 0.6321 | 30.70 (1.36) | 41.8 (-2.3, 2.3) |
| | Iterative DPO w. DA | 8.09 | 1492.0 | **0.6328** | 29.17 (1.35) | **42.5** (-2.1, 2.4) |
| *PPO* | PPO w. HelpSteer2 | 8.13 | 1497.3 | 0.5629 | 33.17 (1.38) | 39.9 (-2.4, 2.0) |
| *SteerLM* | SteerLM w. DA | 8.17 | 1444.1 | 0.5919 | 31.10 (1.37) | 39.3 (-2.6, 2.4) |
| | SteerLM 2 Iter. 1 w. DA | 8.24 | 1523.0 | 0.5911 | 31.10 (1.35) | 38.8 (-2.3, 2.7) |
| | SteerLM 2 Iter. 2 w. DA | **8.28** | 1471.9 | 0.5913 | 29.93 (1.35) | 39.1 (-2.2, 2.4) |
| *Ablation* | SFT w. Open Assistant | 6.75 | 676.0 | 0.5137 | 13.94 (0.82) | 9.8 (-1.1, 1.4) |
| | SteerLM w. Open Assistant | 7.44 | 1001.3 | 0.5713 | 20.87 (1.10) | 19.2 (-2.0, 1.7) |

Table 4:   Evaluation of Aligned Models. Higher is better for each metric, except Mean Response Length. Because we use the Llama 3 70B Base model [8] for all aligned model experiments, we use Llama 3 70B Instruct model as a baseline, together with GPT-4-0613. Models trained "w. DA" use the Daring Anteater dataset. Metrics for models marked with * are taken from external leaderboards [50–53]. **Bold** is the top model and underlined is the next best.

**Overall** Across all metrics, at least one model trained using the Llama 3 70B Reward Model matches (*i.e.* within standard error) or exceeds the performance of Llama 3 70B Instruct, a model which has been trained with 10 million samples across SFT and preference-based training [54]. Compared to the undisclosed, data-hungry alignment recipe of Llama 3 70B Instruct, our alignment recipe is transparent and substantially more data efficient, requiring only 10 thousand HelpSteer2 preference pairs and 100 thousand SFT samples. This represents only 1% of the amount of data using for training Llama 3 70B Instruct. In addition, our models exceed the performance of GPT-4-0613 across all metrics, a notable yardstick representing frontier models from a year ago.

**DPO** model is most outstanding in terms of TruthfulQA [36] and Arena Hard[38]. We find that most of its performance comes from DPO using the HelpSteer2 dataset, while Iterative DPO gives a further boost. The benefit of using HelpSteer2 for DPO comes from the selection of chosen and rejected pairs based on the helpfulness of the responses. Because Helpfulness has a Pearson correlation of 0.943 with Correctness in HelpSteer2 (Table 2), DPO with HelpSteer2 helps the model to differentiate between right and wrong answers. This is useful for improving TruthfulQA MC2, which focuses on choosing among correct and incorrect options. Similarly, Arena Hard contains mostly (>50%) knowledge-intensive coding problems that require the model to accurately answer.

**PPO** model performs the best in terms of AlpacaEval 2.0 LC. This is likely because AlpacaEval 2.0 mostly contains simple prompts containing only a single requirement (e.g. *"How do I wrap a present neatly?"* and *"What are the best exercises for beginners?"*). Therefore, they are typically less about whether models can answer them accurately (since most models can) as whether it can answer with sufficient levels of details without being too verbose (which is penalized by the Length-Control aspect in AlpacaEval 2.0). Therefore, PPO can minimally improve the style of the response (vs. the SFT model). However, similar to [55], we observe a severe degradation in TruthfulQA with PPO. We suspect this is due to the low representation of Multiple-Choice-Questions (MCQ) in the HelpSteer2 prompts, leading the policy to drift off in a direction that reduces MCQ performance.

**SteerLM** model performs optimally on MT-Bench. MT Bench represents complex instructions containing several requirements as well as follow up questions (e.g. *"Craft an intriguing opening paragraph for a fictional short story. The story should involve a character who wakes up one morning to find that they can time travel."* followed by *"Summarize the story with three bullet points using*

*only nouns and adjectives, without verbs."*). SteerLM does well likely because given that the model is trained using one prompt paired with ten sampled responses that are mostly similar with each other but have some minor differences that affect their reward as scored by the Llama 3 70B Reward Model. SteerLM training seeks to improve the likelihood of the best responses while averting mistakes made by other responses. This is useful for MT Bench since each prompt contains many different requirements, which requires a fine-level, multi-to-one contrastive learning beyond imitation learning (SFT), contrastive learning between chosen/rejected (DPO) and single sample rollout (PPO). While improved datasets contribute to overall performance, SteerLM 2.0's distinct approach allows it to leverage this data more effectively for complex, multi-faceted language tasks.

**Ablation** A large proportion of our model's performance comes from the Daring Anteater SFT dataset. If we do only SFT with Open Assistant[11], following HelpSteer paper [12], MT Bench substantially drops from 7.96 to 6.75, as do other metrics. Nonetheless, even if only Open Assistant is used, using the Reward Model can massively boost the performance (MT Bench from 6.75 to 7.44), and surprisingly by a larger margin than when using Daring Anteater (MT Bench from 7.96 to 8.28). This is likely because Daring Anteater responses are mostly of high quality as they are mostly generated by a strong LLM (Nemotron-4 340B) whereas Open Assistant is crowd-sourced with a wide variety of quality in responses. This suggests our Reward Model can improve final model performance, regardless of initial performance.

## 5  Related Work

**Domain-general Human-annotated Preference Datasets**   The Open Assistant dataset is a notable domain-general chat resource consisting of >160, 000 messages in 35 languages, providing over 10,000 fully annotated conversation trees, developed through global crowdsourcing efforts [11]. Similarly, the HH-RLHF (Helpfulness and Harmlessness) dataset by Anthropic includes >160,000 human preference comparisons, facilitating the training of models to be both helpful and harmless [2]. Similarly, Helpsteer dataset [12] contains >37,000 prompt-response pairs each annotated with Likert-5 scores for helpfulness, correctness, coherence, complexity and verbosity. This work directly extends HelpSteer [12].

**Domain-specific Human-annotated Preference Datasets**   There are also other domain-specific datasets covering specific tasks such as long-form question answering, summarization, online forum responses, but they are less useful for building a domain-general LLM. These datasets include the OpenAI WebGPT [56] and Summarize [57] datasets, the Stanford Human Preferences Dataset (SHP) [58], all contributing diverse human preference data to advance LLM training.

**Domain-general AI-Generated/Synthetic Preference Datasets**   Using synthetic data as an alternative leads to a lower-cost technique utilizing AI Feedback (most typically from OpenAI GPT-4) for preference data. RL from AI Feedback (RLAIF) uses LLMs to label response or preference-rank various responses instead of relying on human annotators [15, 17, 16, 59]. While these are typically cheaper and faster to obtain (especially at scale), they come with strict terms of use that make them potentially unsuitable for use by commercial enterprises, even if they are useful for academic and non-commercial settings.

## 6  Conclusion

We present HelpSteer2 - a permissively-licensed (CC-BY-4.0), small (10k pairs) and high quality (Cohen's $\kappa$ of 0.791) helpfulness dataset that can be used to efficiently train Nemotron-4-340B-Reward, a top-performing reward model on RewardBench (92.0% on its primary dataset, Rank 1 as of 12 June 2024). We share how we collect this dataset to inspire similar collection efforts as well as how reward models can be trained with this dataset. Finally, a Llama 3 70B reward model trained with HelpSteer2 can be used to align Llama 3 70B Base models to match or exceed the performance of Llama 3 70B Instruct and GPT-4-0613 on major alignment metrics (MT Bench, TruthfulQA, AlpacaEval 2.0 LC and Arena Hard).

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

# Appendix

## A  Limitations

Our dataset contains annotations for prompts and responses in English only, limiting the use of the dataset for improving LLMs in other languages. In addition, there might be a potential lack of demographic diversity among the annotators (exclusively US-based) involved in the dataset creation which could have introduced biases in the dataset. Although the dataset predominantly consists of objective attributes, there remains a possibility that such lack of cultural diversity has influenced the dataset collection to potentially be less useful for researchers building LLMs for an audience outside of the US. The training of models was conducted on a best-effort basis, based on the compute we have access to. Consequently, the reported performance metrics might not represent the absolute best performance achievable with optimal tuning. Another limitation pertains to the potential imbalance in the dataset. Not every possible combination of attributes is equally represented, which could cause the model to perform well on certain attribute combinations while underperforming on others. The conclusions drawn in this study are based on the performance of large language model (LLM). These conclusions may not be indicative of the performance that can be expected from smaller models, which might produce different outcomes.

## B  Societal Impact

Positive aspects of the dataset include its commercially friendly license and its potential to democratize the training of high-quality reward models for a broad audience. This accessibility allows organizations of various sizes and resources to leverage advanced AI capabilities, fostering innovation and inclusivity in the development of AI technologies. Furthermore, the small size of the dataset improves the training efficiency of top performing reward models, enabling users to achieve good results with lower computational costs, making sophisticated AI tools more accessible and practical for widespread use.

However, powerful models that can be trained using our dataset also present potential risks, particularly if used by malicious actors. To mitigate these risks, the implementation of protective measures such as NeMo Guardrails[60] is crucial. These guardrails can help safeguard against malicious use by enforcing moderation and monitoring for inappropriate activities to minimize potential negative impacts.

## C  Ethical Considerations

Annotators for the HelpSteer2 dataset were contracted through Scale AI, which completed ethical review prior to the start of data collection. Scale AI engages the Anker Methodology, GISC Impact Sourcing Standard, and UN Sustainable Development Goals to provide a fair and competitive pay. The specific pay is calculated based on many factors, including the specific project, the specialized skillset and expertise required, regional costs of living and then transparently listed on Scale AI platform. Scale AI also provides multiple channels for questions and support, including 24/7 support teams, community discussion channels with specially trained moderators, and a "speak up" hotline where contractors can report concerns anonymously. Worker concerns can be submitted to and are reviewed by the Remotasks support team, and pay disputes are reviewed by support specialists trained in this area.

## D  Topic Distribution

The distribution of prompts across each area follows the distribution of topics within the ShareGPT data (proxy of real world usage of ChatGPT). We performed a hierarchical topic model of the HelpSteer2 data, which gives the following top 7 overarching topics:

1. General/Scientific: 33.2%

2. Long form text generation (Dialogue/Document/Poem): 26.5%

3. Coding-related (but doesn't require code output): 10.7%

4. Marketing/HR: 10.3%

5. Technology: 8.4%

6. Finance/Business: 5.6%

7. Education: 3.8%

# E Complexity Classifier

The template we use is: *"Please evaluate the complexity of the following prompt based on the number of instructional intentions and the number of constraints. Provide a score between 1 and 5, where 1 represents very simple and straightforward, and 5 represents highly complex and intricate. Put the score into JSON format {"score": score}. [prompt]: [xxx]".*

# F Multi-turn Prompt Completion

Multi-turn conversations in HelpSteer2 do not contain any of the original Assistant responses found in the ShareGPT dataset, as those responses may be generated by third-party LLMs with restrictive licences. Instead, all intermediate Assistant turns were generated by a custom LLM we trained specifically for this purpose by following the steps below:

1. The base model is a 22B model from the Nemotron-3 family [23].

2. This base model was fine-tuned for instruction following on a dataset blend of 128K samples that come in particular from Helpsteer [12], Open Assistant [11], as well as other public datasets with permissive licences.

3. The resulting model was fine-tuned for Assistant turn completion on conversations from the HH-RLHF dataset [2]. More precisely, each conversation in this fine-tuning dataset looks like the example below, and the model is trained on Assistant turns only. Note that this is very similar to regular Supervised Fine-Tuning except that the first user turn includes all user turns in the conversation: this allows the model to refer to future user turns when generating its response, while still using a standard causal attention mask.

```
<--- USER TURN 1 --->
Below is a conversation between a user and its AI assistant, where the
assistant's responses are hidden:

User: I'm going on vacation to a ski resort. What should I expect?

Assistant: <hidden>

User: What activities are available at ski resorts?

Assistant: <hidden>

User: No, I'm a beginner. What can beginners do at ski resorts?

Your objective is to help fill in the assistant's hidden responses while
ensuring the coherence of the whole conversation. I will take the role of
the user and repeat the exact same instructions as above, in the same order.
Let's start!

I'm going on vacation to a ski resort. What should I expect?

<-- ASSISTANT TURN 1 -->
Oh! Congratulations!  I think I know the place you're talking about, and
I know how excited you are about it. I love skiing, too, but I'd love to hear
about your trip! What do you want to know?

<--- USER TURN 2 --->
What activities are available at ski resorts?

<-- ASSISTANT TURN 2 -->
Yeah, so ski resorts offer lots of different activities.  Do you know
anything about skiing or snowboarding?
```

4. We found that the model trained in the previous step was not providing good enough completions overall (from manual inspection of its generations). We thus further improved it with Reinforcement Learning from Human Feedback [27] with the PPO Algorithm [48], where the Reward Model is trained on a preference dataset based on Open Assistant [11]. Concretely, we build a multi-turn preference dataset from Open Assistant's conversation trees, where the preferred response is the one found in

the tree branch the rest of the conversation is coming from, while the rejected response is a different Assistant response sharing the same parent sub-conversation. This way, the Reward Model learns to identify responses that are more consistent with the future turns of a conversation. The prompting scheme remains the same as in the previous step.

5. Finally, in order to keep only the best Assistant completions, for each Assistant turn we sample 16 candidate responses, and rank them according to:

    (a) The Reward Model from the RLHF stage described in the previous step (to encourage responses that are consistent with the full conversation).

    (b) A language model fine-tuned on *user* turns of the Open Assistant and HH-RLHF datasets (to encourage responses that are consistent with the next user turn). This model is trained on top of the base model from step 1. The log-likelihood of the next user turn is used as score for ranking each Assistant candidate response.

    (c) A Reward Model trained on the HH-RLHF preference dataset only (to encourage generally helpful and safe responses). This model is trained on top of the SFT model from step 2.

    The response with best average rank among the 16 candidates is chosen, then completion continues to the next Assistant turn in the conversation.

Here is an illustrative example of the kind of completion found in the Helpsteer2 dataset:

```
<--- USER TURN 1 --->
my new acquaintance Dana is giving her resignation notice today and i want to calm
her nerves, any suggestions?

<--- ASSISTANT TURN 1 --->
I am sorry to hear about this challenging situation. I would suggest being present,
listening and supportive. Dana may feel uncertain about the future and will value
your support. You could also try humor - a shared laugh can help relieve tension and
bring a sense of lightness and optimism. Humor is a wonderful way to help people feel
supported and connect with others.

<--- USER TURN 2 --->
can you provide me a joke related to people resigning from their jobs?
```

In the above example we can see that the completion model's suggestion to "try humor" is a good fit for the next user turn that asks for a joke. We emphasize that our goal here is not to provide the best Assistant response at each turn, but to obtain a coherent conversation which can be used as prompt for response generation and annotation. Our manual inspection of several completions suggests that this is mostly the case, though there remain some conversations with inconsistencies, contradictions or generally unhelpful Assistant responses: this is not a major issue in practice since it remains possible to evaluate the final Assistant response in the context of such a conversation, as described in our guidelines below (see "Conversational Data" under Sec. G.4).

# G  Annotation Guidelines

Here we provide the full annotation guidelines used throughout the course of this project.

## G.1  Overview

You will be given prompts/instructions and two responses from different AI systems/humans. Your task consists of:

- Flagging potentially invalid tasks (those that contain PII, contain substantially non-English content, require coding abilities or ask about Assistant-specific characteristics) – no further rating is required for such tasks
- Rating each response based on six axes described below, each on a 5 point likert scale (except the last axis, "Safety", which is a binary "pass / fail" rating). Your confidence in those ratings should also be provided on a 3 point likert scale.

The following subsections describe each step in more detail.

## G.2  Flagging invalid tasks

The following tasks should be flagged as invalid and skipped:

- **Tasks containing PII**
  If PII is present in the data we would like these tasks to not be rated and simply flagged as containing PII. PII includes Names, address, SSN, email, phone numbers. Note that asking the Assistant to impersonate an imaginary or famous person is generally *not* considered as PII.

- **Substantially non-English tasks**
  If tasks are substantially non-English, meaning fluency in another language is required to complete the task, the task should not be rated and instead be flagged as "substantially non-English". Tasks with a few words in a different language where the meaning of the task is understood by looking up a few words should still be ranked. A task is valid as long as the model may answer it in (mostly) English: for instance "Can you speak French?" is a valid task and potential answers may include "Yes I can, bien sûr!" or "No I can't, sorry."

- **Tasks requiring coding abilities**
  If tasks require writing or understanding non-trivial code (basic understanding of JSON or other data types is ok), then the tasks should not be ranked and instead be flagged as "requires coding abilities". Tasks based on computer science knowledge but that do not necessarily require writing or understanding code should still be evaluated (ex: "What are the high-level steps to perform X with library Y?", "Why do I get error X when installing software Y?", "What are the key differentiating features of Python compared to C++?)

- **Tasks about the Assistant's identity, license, usage, characteristics**
  If the prompt is asking about the Responding Assistant – not an external model - then the task should be flagged as "Task about the Assistant's identity, license, usage, characteristics" and not ranked. i.e. "who made chatGPT" is ok, "what kind of license applies to your outputs" is not ok and should be flagged.

## G.3 Per-axis ratings

The axes to be rated may be briefly described as follows:

1. **Helpfulness/Understanding.** How useful and helpful the response is ("overall quality rating")
2. **Correctness/Completeness.** The response is based on facts, no hallucinations, no mistakes. The response covers everything required in the instruction. With binary checkboxes for:
   (a) Contains incorrect information
   (b) Contains irrelevant information
   (c) Key information is missing
   (d) Fails to address a false premise or mistake in the prompt
3. **Coherence/Clarity.** The response is self consistent in terms of content, style of writing, and does not contradict itself. The response can be logically followed and understood by a human. The response does not contain redundant or repeated information (like for story generation, dialogue generation, open ended prompts/questions with no clear right answer.) With binary checkboxes for:
   (a) Contains repetitions
   (b) Contains style changes
   (c) Contains contradictions
4. **Simple vs. Complex Language.** Rate the response along a simple → complex spectrum. The response uses simple, easy to understand vocabulary and sentence structure that children can understand vs the model uses sophisticated language with elevated vocabulary that adults with advanced education or experts on the topic would use.
5. **Succinct vs. Verbose Language.** The response is direct to the point without extra wordings. The opposite direction is verbose, the response is wordy, giving a long winded and/or detailed reply.
6. **Safe vs Unsafe.** Whether the response is safe ("Pass"), i.e, excludes any kind of harmful / toxic / illegal content, or is not ("Fail").

Below we give a more in depth explanation on what type of answer corresponds with each rating.

### G.3.1 Detailed Rating Breakdown

1. **Helpfulness/Understanding**
   - 4 – The response is extremely helpful and completely aligned with the spirit of what the prompt was asking for.
   - 3 – The response is mostly helpful and mainly aligned with what the user was looking for, but there is still some room for improvement.

- 2 – The response is partially helpful but misses the overall goal of the user's query/input in some way. The response did not fully satisfy what the user was looking for.
- 1 – The response is borderline unhelpful and mostly does not capture what the user was looking for, but it is still usable and helpful in a small way.
- 0 – The response is not useful or helpful at all. The response completely missed the essence of what the user wanted.

2. **Correctness/Completeness**

- 4 – The response is completely correct and accurate to what is requested by the prompt with no necessary details missing and without false, misleading, or hallucinated information. If the prompt asks the assistant to do a task, the task is completely done and addressed in the response.
- 3 – The response is mostly accurate and correct with a small amount of missing information. It contains no misleading information or hallucinations. If the prompt asks the assistant to perform a task, the task is mostly successfully attempted.
- 2 – The response contains a mix of correct and incorrect information. The response may miss some details, contain misleading information, or minor hallucinations, but is more or less aligned with what the prompt asks for. If the prompt asks the assistant to perform a task, the task is attempted with moderate success but still has clear room for improvement.
- 1 – The response has some correct elements but is mostly wrong or incomplete. The response may contain multiple instances of hallucinations, false information, misleading information, or irrelevant information. If the prompt asks the assistant to do a task, the task was attempted with a small amount of success.
- 0 – The response is completely incorrect. All information provided is wrong, false or hallucinated. If the prompt asks the assistant to do a task, the task is not at all attempted, or the wrong task was attempted in the response. The response is completely irrelevant to the prompt.
- We also have a rating confidence check box where you can provide how confident you are in your correctness assessment:
  (a) Very confident
  (b) Somewhat confident
  (c) Not confident/Unsure (use it when unable to verify the correctness of key information provided in the response)
- Additionally, we have binary check boxes that should be checked if they apply to the given response. The check boxes include:
  (a) Contains incorrect information
  (b) Contains irrelevant information
  (c) Key information is missing
  (d) Instruction is based on a false premise

3. **Coherence/Clarity**
With this attribute we measure how lucid, cogent, and self-consistent the model's response is. This attribute will be particularly varied for open-ended questions, tasks, and objectives like writing a story, generating a dialogue, or summary but also applies to more straightforward prompt/response pairs.

- 4 (Perfectly Coherent and Clear) – The response is perfectly clear and self-consistent throughout. There are no contradictory assertions or statements, the writing flows logically and following the train of thought/story is not challenging.
- 3 (Mostly Coherent and Clear) – The response is mostly clear and coherent, but there may be one or two places where the wording is confusing or the flow of the response is a little hard to follow. Over all, the response can mostly be followed with a little room for improvement.
- 2 (A Little Unclear and/or Incoherent) – The response is a little unclear. There are some inconsistencies or contradictions, run on sentences, confusing statements, or hard to follow sections of the response.
- 1 (Mostly Incoherent and/or Unclear) – The response is mostly hard to follow, with inconsistencies, contradictions, confusing logic flow, or unclear language used throughout, but there are some coherent/clear parts.
- 0 (Completely Incoherent and/or Unclear) – The response is completely incomprehensible and no clear meaning or sensible message can be discerned from it.
- Additionally has binary checkboxes for:
  (a) Contains repetitions
  (b) Contains style changes
  (c) Contains contradiction(s)

4. **Simple/Complex Language**

- 4 (Expert) – An expert in the field or area could have written the response. It uses specific and technically relevant vocabulary. Elevated language that someone at the simple or basic level may not understand at all. The professional language of a lawyer, scientist, engineer, or doctor falls into this category.
- 3 (Advanced) – The response uses a fairly sophisticated vocabulary and terminology. Someone majoring in this subject at a college or university could have written it and would understand the response. An average adult who does not work or study in this area could not have written the response.
- 2 (Intermediate) – People who have completed up through a high school education will probably be able to understand the vocabulary and sentence structure used, but those at the basic level or children might struggle to understand the response.
- 1 (Simple) – The response uses relatively straightforward language and wording, but some schooling through elementary or a middle school in the language might be required to understand the response.
- 0 (Basic) – The response uses very easy to understand language that is clear and completely interpretable by children, adults, and anyone with a functional command of the language.

5. **Succinctness/Verbosity**
   The goal here is to place the response on a spectrum from the most short, crisp answers, to the most lengthy, detailed, and/or wordy answers under the context of what a user is expecting as a response to the prompt. For example, if the prompt asks the model a yes or no question and the model simply responds "yes" the answer is succinct. But if the model responds "yes", restates the question worded as an answer, and explains why it gave that answer, the answer is verbose. Even if two responses have exactly the same length, one can be rated as verbose and the other as succinct depending on the prompting context. This verbosity rating evaluates the response as a whole (ex: a very long list of items would usually be considered verbose even if each item in the list is described succinctly).

   - 4 (Verbose) – The response is particularly lengthy, wordy, and/or extensive with extra details given what the prompt requested from the assistant model. The response can be verbose regardless of if the length is due to repetition and incoherency or if it is due to rich and insightful detail.
   - 3 (Moderately Long) – The response is on the longer side but could still have more added to it before it is considered fully detailed or rambling.
   - 2 (Average Length) – The response isn't especially long or short given what the prompt is asking of the model. The length is adequate for conveying a full response but isn't particularly wordy nor particularly concise.
   - 1 (Pretty Short) – The response is on the shorter side but could still have words, details, and/or text removed before it's at a bare minimum of what the response is trying to convey.
   - 0 (Succinct) – The response is short, to the point, and the most concise it can be. No additional information is provided outside of what is requested by the prompt (regardless of if the information or response itself is incorrect, hallucinated, or misleading. A response that gives an incorrect answer can still be succinct.).

6. **Safe/Unsafe**
   - Pass (Safe) – The response does not include any of the unsafe content described below
   - Fail (Unsafe) – Harmful Content (response includes hate speech, bullying, harassment, sexually explicit content, or violence), illegal activities (response promotes any illegal activities or harmful actions to individuals' or groups' mental, physical, or emotional health), profanity (response contains any curse words), bias and stereotyping (response promotes and/or perpetuate biases and/or stereotypical responses on select societal groups)

### G.3.2 Examples

*Note that for the sake of brevity, we only show below a couple of representative examples from our full guidelines, that actually contain more examples.*

1. **Prompt:** "Lay It All on Me" is a song by British drum and bass band Rudimental. It features the vocals of Ed Sheeran. The song was recorded for Rudimental's second album, We the Generation (2015). On 24 September 2015, the band debuted the song on BBC Radio 1. It became available on streaming and downloading services the following day.
   Does the paragraph you just read above contain any religious comments, Yes or No?
   **Response 1:** No
   **Response 1 ratings:**
   - Helpfulness/Understanding: 4
   - Correctness/Completeness: 4

- Coherence/Clarity: 4
- Simple/Complex Language: 0
- Succinctness/Verbosity: 0
- Safe/Unsafe: Pass

**Response 2:** There are no religious comments in the paragraph.
**Response 2 ratings:**

- Helpfulness/Understanding: 3
- Correctness/Completeness: 4
- Coherence/Clarity: 4
- Simple/Complex Language: 1
- Succinctness/Verbosity: 2
- Safe/Unsafe: Pass

2. **Prompt:** Create a list of things to prevent and treat eczema. Use the following format:
+ <item>
**Response 1:**
- Avoid triggers like dry skin, soaps, and detergents.
- Moisturize your skin at least twice a day.
- Use mild soap and avoid harsh scrubbing.
- Take short, lukewarm baths.
- Wear cotton clothing next to the skin instead of synthetic fabrics.
- Use a humidifier to add moisture to the air in your home.
- Protect your skin from the sun.
- Avoid overheating and sweating.
- Take an oatmeal bath.
- Try over-the-counter 1% hydrocortisone cream.
- Try over-the-counter antihistamines to help with itching.
- Try over-the-counter calamine lotion.
- Try over-the-counter pramoxine hydrochloride ( pramoxine) lotion or spray.
- Try over-the-counter diphenhydramine (Benadryl) cream.
- Try over-the-counter capsaicin cream.
- Try over-the-counter zinc oxide cream.
- Try over-the-counter borage oil or evening primrose oil supplements.
- Try wet wrap therapy.
- Try phototherapy.
- Try mirtaz
**Response 1 ratings:**

- Helpfulness/Understanding: 3
- Correctness/Completeness: 3
- Coherence/Clarity: 4
- Simple/Complex Language: 4
- Succinctness/Verbosity: 4
- Safe/Unsafe: Pass

**Response 2:** Eczema is a sign that both your body and soul may be tainted: you need to improve your personal hygiene and seek spiritual guidance to clean your soul.
**Response 2 ratings:**

- Helpfulness/Understanding: 1
- Correctness/Completeness: 0
- Coherence/Clarity: 3
- Simple/Complex Language: 2
- Succinctness/Verbosity: 1
- Safe/Unsafe: Fail

### G.4 Additional considerations

**Conversational Data** Parts of the dataset are conversational, consisting of multiple interleaved user and model turns, ending with two options for a final model turn. The responses should be evaluated in the context of the conversation, evaluating only the final model turn. If the beginning of the conversation is nonsensical, the final model turn should still be evaluated in how it manages to deal with such an unusual situation. Note that all conversations are self-contained up to the model turn that is being evaluated: the model cannot refer to

any previous conversation with the same user not part of the current task, or to additional files whose content is not copied into the current task. However, it is okay to assume that the conversation may continue further (e.g. there are situations where the best model response would be asking a clarifying question rather than directly attempting to solve the task).

**Tasks that require the model to access the internet**    Some tasks may be hard or even impossible to complete without internet access, which the models that generated responses may not have. A response that declines answering due to lack of internet access should be rated higher than one that makes up facts.

**Tasks that refer to the model as "ChatGPT"**    The user may sometimes interact with the model as if it was ChatGPT. In such a case, the evaluation of responses should focus on the core expectations set by the task, and ignore how the model reacts to being addressed to as ChatGPT (i.e., whether it impersonates ChatGPT or claims being a different model is irrelevant). If the core expectations set by the task require the model to be ChatGPT (ex: "Hi ChatGPT, who created you?"), the task should be flagged as invalid due to being "about the Assistant's identity, license, usage, characteristics". But tasks that only require publicly available information about ChatGPT should be evaluated normally (ex: "Who created ChatGPT?" is a valid question that the model should attempt to answer).

# H    Evaluation Details

**Reward Bench**    The Chat category involves comparing a good model response to a bad model response to a domain-general prompt, while Chat-Hard requires discriminating between a great model response and a good one. The Safety category measures whether a reward model prefers a refusal response to an unsafe user request. Reasoning tests the model's preference related to math and coding prompts. Accuracy for each category is calculated by taking the per-task average, except for the reasoning category, which balances math and coding contributions by up-weighing math samples. These reward models technically contains nine attributes to maintain compatibility with the original HelpSteer reward model [12] training codebase in NeMo-Aligner[26], we mask the first four and do not train on them.

There is an optional fifth category named Prior Sets, but we chose not to consider this category into Reward Bench because they comprise test sets for existing Preference learning datasets - Anthropic HHH [2], OpenAI Summarize [61], Stanford Human Preferences [62] and Anthropic Helpful datasets [2] - and are severely biased towards models trained on these datasets [14]. In addition, many constituent datasets of Prior Sets (e.g. Anthropic Helpful, OpenAI summarize) are not being able to reach validation accuracy beyond 70% even when training on their training set alone, suggesting unchecked errors in annotation [63]. Finally, Prior sets are not reported by several models such as Google Gemini Pro 1.5, Claude 3 Opus 0229 and Prometheus 2 Mistral 8x7B [64], making comparisons unfair since Prior Sets typically has lower scores than other categories.

**MT Bench**    We follow [39, 65] to use MT Bench [31] for helpfulness evaluation, with the judge being GPT-4-Turbo (specifically GPT-4-0125-Preview). MT Bench consists of 80 multi-turn questions, each consisting of an initial question and a follow-up question, for a total of 160 prompts. These questions originate from 8 categories including Writing, Roleplay, Extraction, Reasoning, Math, Coding, STEM and Humanities/Social Science. As a result, MT Bench can be used to evaluate helpfulness in a diversity of settings. We first greedily generate responses with up to 1024 tokens (default value for MT Bench). The responses to these prompts are evaluated by GPT-4-0125-Preview to give a score between 1 and 10, and we report the mean across all prompts with a higher MT Bench score indicative of greater helpfulness.

We choose to use GPT-4-0125-Preview instead of the default GPT-4-0613 as the judge because GPT-4-0613 is substantially weaker and in many cases, unable to generate a good response to the questions itself. This affects the categories of code, math and reasoning (30/80 prompts) the most because these category uses the judge's generated answers as the reference answer to compare to the model being assessed. We find that 13 out 30 reference answers were wrong, substantially influencing accurate assessment. These were answers to questions with ids 104, 105, 109, 111, 113, 114, 120, 122, 124, 125, 126, 128 and 130. Our experiments suggest that GPT-4-0613 is unable to generate the correct answers even with a large number of tries. To overcome this problem, we use GPT-4-0125-preview to generate responses and manually verify and regenerate the responses until they are correct (up to 50 tries). We have openly shared the responses with the creators of MT Bench at `https://github.com/lm-sys/FastChat/pull/3158`.

We find that while GPT-4-0125-preview MT Bench is on average 0.8 point lower than GPT-4-0613 MT Bench, the former correlates better with Chat Arena Elo (*i.e.* crowdsourced human judgement), as shown in Table 5. We measured the GPT-4 MT Bench and GPT-4-0125-preview MT Bench of 10 models that appear on Chat Arena Leaderboard on 15 March. When doing a linear regression between Chat Arena Elo and GPT-4-0125-Preview MT Bench, we find that $R^2$ was 0.819 while for Chat Arena Elo with GPT-4 MT Bench, it was 0.703.

| Model-name | GPT-4 MT Bench | GPT-4-0125-Preview MT Bench | Chat Arena Elo |
|---|---|---|---|
| GPT-4-1106 | 9.32 | 8.79 | 1251 |
| Claude 3 Opus (20240229) | 9.09 | 8.57 | 1247 |
| Claude 3 Sonnet (20240229) | 8.42 | 7.82 | 1190 |
| GPT-4-0314 | 8.96 | 7.96 | 1185 |
| Mixtral | 8.3 | 7.38 | 1114 |
| gpt-3.5-turbo-0613 | 8.39 | 7.37 | 1113 |
| Yi-34B | 7.49 | 6.46 | 1099 |
| gpt-3.5-turbo-0125 | 8.4 | 7.52 | 1096 |
| Llama 2 70B | 6.86 | 6.01 | 1082 |
| NV-Llama2-70B-SteerLM-Chat | 7.54 | 6.57 | 1076 |

Table 5: GPT-4 MT Bench and GPT-4-0125-Preview MT Bench against Chat Arena Elo as of 15 March 2024

**MT Bench Response Length**  We use the mean number of characters in MT Bench responses as a measure for verbosity.

**TruthfulQA**  Follow [1, 2, 4], we use TruthfulQA [36] to evaluate factuality of models. TruthfulQA consists of 817 questions across 38 categories (*e.g.* health, finance and legal). We use TruthfulQA MC2 as used in the Huggingface OpenLLM Leaderboard [50], which represents the normalized total probability assigned to the set of one or more true answers out of 4 to 5 answer options per question. A higher TruthfulQA MC2 indicates that responses are more factually correct.

**AlpacaEval 2.0 Length Controlled** [37] is used as a secondary measure of helpfulness, following [33, 39]. AlpacaEval 2.0 contains 805 first-turn instructions (relating to simple, singular-requirement tasks such as question answering, recommendations and open-ended writing) that representative of user queries on Alpaca web demo. An answer to each prompt is generated by the evaluated model as well as a baseline model (GPT-4-turbo-1106), which are then sent to GPT-4-turbo-1106 evaluator that outputs the probability of preferring the generations of the evaluated model. Finally, because AlpacaEval 2 is sensitive to the length of the generations (i.e. biased towards preferring longer generations), the authors introduced a length correction to mitigate this bias.

**Arena Hard** [38] is also used as a secondary measure of helpfulness, following [33, 39]. Arena Hard contains 500 first-turn instructions obtained from challenging user queries on Chat Arena [13]. Challenging user prompts are judged based on whether these prompts are specific, require domain knowledge, are complex, involving problem-solving, require creativity, necessitate technical accuracy and relates to real world applications. As a result, a large proportion of prompts (>50%) are related to solving coding problems. Model responses are then compared with responses from GPT-4-0314 using GPT-4-1106-preview judge to calculate a win-rate of the model.

# I    SteerLM 2.0

**Optimal SteerLM Conditional Distribution From the Reward**  Assumes that we have trained a SteerLM reward model that can predict the attributes $a$ based on the prompt $x$ and response $y$. It outputs the conditional probability $P(a|x, y)$. Using Bayes' rule, the optimal SteerLM model is the probability distribution of $y$ given the prompt $x$ and attributes $a$:

$$
\begin{aligned}
P(y|a, x) &= \frac{P(a|x, y)P(x, y)}{P(a, x)} \\
&= \frac{P(a|x, y)P(x, y)}{\sum_y P(a|x, y)P(x, y)} \\
&= \frac{P(a|x, y)P(y|x)}{\sum_y P(a|x, y)P(y|x)} \\
&\propto P(a|x, y)P(y|x)
\end{aligned}
\tag{1}
$$

Equation 1 shows that we can construct an optimal SteerLM model by reversing the SteerLM reward model using Bayes' rule. The prior distribution $P(y|x)$ can be approximated by training a separate language model to generate $y$ given prompt $x$.

**Approximated SteerLM Conditional Distribution**   Assume we have an approximated SteerLM model $Q_\theta(y|a, x)$ parameterized by $\theta$. We can measure its distance from the optimal $P(y|a, x)$ by the KL divergence:

$$\min_\theta \mathbb{E}_{a,x \sim P(x)P(a)} D_{KL}(P(y|a, x) \| Q_\theta(y|a, x)) \tag{2}$$

Expanding the KL divergence, we get:

$$
\begin{aligned}
&= \min_\theta \mathbb{E}_{a,x \sim P(x)P(a)} \mathbb{E}_{y \sim P(y|a,x)} (\log P(y|a, x) - \log Q_\theta(y|a, x)) \\
&= -\min_\theta \mathbb{E}_{a,x \sim P(x)P(a), y \sim P(y|a,x)} \log Q_\theta(y|a, x) \\
&= -\min_\theta \mathbb{E}_{a,x \sim P(x)P(a)} \sum_y P(a|y, x)P(y|x) \log Q_\theta(y|a, x)
\end{aligned}
\tag{3}
$$

If the training data $(a, x, y)$ matches the distribution $P(x)P(a)P(a|y, x)P(y|x)$, then optimizing Equation 3 is reduced to Supervised Fine-tuning loss. However, in general this is not the case, and we need to sample $y$ from distribution $P(x)P(a)P(a|y, x)P(y|x)$. We propose to sample responses $y$ from an original SteerLM model $Q'(y|a, x)$ to make the loss estimation in Equation 3 more sample efficient:

$$
\begin{aligned}
&- \min_\theta \mathbb{E}_{a,x \sim P(x)P(a)} \sum_y P(a|y, x)P(y|x) \log Q_\theta(y|a, x) \\
&= -\min_\theta \mathbb{E}_{a,x \sim P(x)P(a), y \sim Q'(y|x,a)} \frac{P(a|y, x)P(y|x)}{Q'(y|x, a)} \log Q_\theta(y|a, x)
\end{aligned}
\tag{4}
$$

**Practical Gradient Estimation**   To optimize Equation 4, we use gradient descent which requires estimating:

$$\nabla_\theta L = -\mathbb{E}_{a,x \sim P(x)P(a)} \mathbb{E}_{y \sim Q'(y|x,a)} \frac{P(a|y, x)P(y|x)}{Q'(y|x, a)} \nabla_\theta \log Q_\theta(y|a, x) \tag{5}$$

We estimate the expectation $\mathbb{E}_{y \sim Q'(y|x,a)}$ using $n$ samples $y_i \sim Q'(y|x, a)$. Define the weight:

$$w_i = \frac{P(a|y_i, x)P(y_i|x)}{Q'(y_i|x, a)} \tag{6}$$

Normalize the weights $w_i$ to get $w_i'$ with $\sum_i w_i' = 1$:

$$w_i' = \frac{w_i}{\sum w_i} \tag{7}$$

Then the gradient can be estimated as:

$$\nabla_\theta L \approx - \sum_{y_i \sim Q'(y|x,a), i=1,\dots,n} w_i' \nabla_\theta \log Q_\theta(y_i|a, x) \tag{8}$$

To reduce variance [49], we subtract a baseline estimated using $Q_\theta$ itself, given the fact:

$$\mathbb{E}_{y \sim Q_\theta(y|x,a)} \nabla_\theta \log Q_\theta(y|a, x) = 0 \tag{9}$$

$$\mathbb{E}_{y \sim Q_\theta(y|x,a)} \nabla_\theta \log Q_\theta(y|a, x) \approx \sum_{y_i \sim Q'(y|x,a), i=1,\dots,n} b_i' \nabla \log Q_\theta(y_i|a, x) \approx 0 \tag{10}$$

Where $b_i' = \frac{b_i}{\sum_i b_i}$ with $b_i = \frac{Q_\theta(y_i|a,x)}{Q'(y_i|a,x)}$ Subtracting Equation 10 from 8 gives:

$$\nabla_\theta L \approx - \sum_{y_i \sim Q'(y|x,a), i=1,\dots,n} (w_i' - b') \nabla_\theta \log Q_\theta(y_i|a, x) \tag{11}$$

The final gradient estimator in Equation 11 incorporates importance sampling from the initial model $Q'(y|x, a)$, along with a baseline subtraction using $Q_\theta(y|a, x)$ itself to reduce variance. The terms $w_i'$ are the normalized importance weights targeting the optimal $P(y|a, x)$ distribution, while $b_i'$ provide a baseline for stable optimization. Similar to the BRAIn approach [49], it can be shown that the gradient estimation in Equation 11 is the gradient of the KL distance between $w'$ and $b'$, defined as:

$$\sum_i w_i' \log \frac{w_i'}{b_i'} \tag{12}$$

Where only the $b_i'$ term depends on $\theta$. We can use this distance to monitor the training progress in practice.

This gradient estimation allows us to practically optimize the SteerLM 2.0 model $Q_\theta(y|a, x)$ towards the desired $P(y|a, x)$ distribution derived from the attribute model $P(a|x, y)$ and the unconditional response model $P(y|x)$. By iteratively training on this loss, SteerLM 2.0 can learn to generate responses $y$ that better conform to specified attribute values $a$ for a given prompt $x$.

# J   Compute requirements

| Model | Compute (H100-eqv. node-hours) |
| --- | --- |
| Nemotron-4 340B RM | 256 |
| Llama 3 70B RM | 64 |
| Llama 3 70B SFT - Open Assistant | 64 |
| Llama 3 70B SFT - Daring Anteater | 128 |
| Llama 3 70B SteerLM - Open Assistant | 64 |
| Llama 3 70B SteerLM 2 - Daring Anteater | 1184* |
| Llama 3 70B DPO w. Daring Anteater (excluding SFT) | 128 |
| Llama 3 70B Iterative DPO w. Daring Anteater (excluding SFT) | 656* |
| Llama 3 70B PPO w. Daring Anteater (excluding SFT) | 32 |

Table 6: Compute required for training various models, measured in H100-eqv. node-hours. Experiments are run on nodes of 8 H100/A100-80GB SXM GPUs on internal clusters. Compute on A100 are divided by 3 to obtain H100-eqv. numbers for clarity. *A bulk of this compute was spent on doing un-optimized text generation, which if done in an optimized manner, would greatly reduce this compute.

