# K Supplementary Materials

Include extra information in the appendix. This section will often be part of the supplemental material. Please see the call on the NeurIPS website for links to additional guides on dataset publication.

1. Submission introducing new datasets must include the following in the supplementary materials:

    (a) Dataset documentation and intended uses. Recommended documentation frameworks include datasheets for datasets, dataset nutrition labels, data statements for NLP, and accountability frameworks. **See attached**

    (b) URL to website/platform where the dataset/benchmark can be viewed and downloaded by the reviewers. **https://huggingface.co/datasets/nvidia/HelpSteer2**

    (c) URL to Croissant metadata record documenting the dataset/benchmark available for viewing and downloading by the reviewers. You can create your Croissant metadata using e.g. the Python library available here: https://github.com/mlcommons/croissant **https://huggingface.co/api/datasets/nvidia/HelpSteer2/croissant**

    (d) Author statement that they bear all responsibility in case of violation of rights, etc., and confirmation of the data license. **The authors bear all responsibility in case of violation of rights and confirm the data license as CC-BY-4.0.**

    (e) Hosting, licensing, and maintenance plan. The choice of hosting platform is yours, as long as you ensure access to the data (possibly through a curated interface) and will provide the necessary maintenance. **We plan to host and maintain it on Huggingface at the URL above**

2. To ensure accessibility, the supplementary materials for datasets must include the following:

    (a) Links to access the dataset and its metadata. This can be hidden upon submission if the dataset is not yet publicly available but must be added in the camera-ready version. In select cases, e.g when the data can only be released at a later date, this can be added afterward. Simulation environments should link to (open source) code repositories. **https://huggingface.co/datasets/nvidia/HelpSteer2**

    (b) The dataset itself should ideally use an open and widely used data format. Provide a detailed explanation on how the dataset can be read. For simulation environments, use existing frameworks or explain how they can be used. **It's in jsonlines format for machine-readability and has a visual user-interface on Huggingface to be human-readable without needing code.**

    (c) Long-term preservation: It must be clear that the dataset will be available for a long time, either by uploading to a data repository or by explaining how the authors themselves will ensure this. **We plan to host and maintain it on Huggingface at the URL above**

    (d) Explicit license: Authors must choose a license, ideally a CC license for datasets, or an open source license for code (e.g. RL environments). **We use CC-BY-4.0 license**

    (e) Add structured metadata to a dataset's meta-data page using Web standards (like schema.org and DCAT): This allows it to be discovered and organized by anyone. If you use an existing data repository, this is often done automatically. **https://huggingface.co/api/datasets/nvidia/HelpSteer2/croissant**

    (f) Highly recommended: a persistent dereferenceable identifier (e.g. a DOI minted by a data repository or a prefix on identifiers.org) for datasets, or a code repository (e.g. GitHub, GitLab,...) for code. If this is not possible or useful, please explain why. **It's not clear how we can do it with a Huggingface dataset repository**

3. For benchmarks, the supplementary materials must ensure that all results are easily reproducible. Where possible, use a reproducibility framework such as the ML reproducibility checklist, or otherwise guarantee that all results can be easily reproduced, i.e. all necessary datasets, code, and evaluation procedures must be accessible and documented.

4. For papers introducing best practices in creating or curating datasets and benchmarks, the above supplementary materials are not required.

# DataSheet for HelpSteer2 Dataset

## I. MOTIVATION FOR DATASHEET CREATION

### A. Why was the datasheet created? (e.g., was there a specific task in mind? was there a specific gap that needed to be filled?)

We wanted to have a high quality, permissively-licensed helpfulness dataset available to the community for model alignment for this wasn't yet available.

### B. Has the dataset been used already? If so, where are the results so others can compare (e.g., links to published papers)?

The only use of the dataset so far lies in our attempts to do model alignment in the submitted paper.

### C. What (other) tasks could the dataset be used for?

Model Alignment - i.e. training base LLMs to become helpful assistants

### D. Who funded the creation dataset?

NVIDIA

### E. Any other comment?

No.

## II. DATASHEET COMPOSITION

### A. What are the instances?(that is, examples; e.g., documents, images, people, countries) Are there multiple types of instances? (e.g., movies, users, ratings; people, interactions between them; nodes, edges)

Text

### B. How many instances are there in total (of each type, if appropriate)?

20324 train and 1038 validation.
21362 total.

### C. What data does each instance consist of ? "Raw" data (e.g., unprocessed text or images)? Features/attributes? Is there a label/target associated with instances? If the instances related to people, are subpopulations identified (e.g., by age, gender, etc.) and what is their distribution?

Each instance comes with a prompt to an LLM and its respone as well as human labels for 5 attributes (helpfulness, correctness, coherence, complexity and verbosity). Each label is a likert-5 label (between 0 and 4).

### D. Is there a label or target associated with each instance? If so, please provide a description.

Yes - each sample has an annotated response with human labels for 5 attributes (helpfulness, correctness, coherence, complexity and verbosity). Each label is a likert-5 label (between 0 and 4).

### E. Is any information missing from individual instances? If so, please provide a description, explaining why this information is missing (e.g., because it was unavailable). This does not include intentionally removed information, but might include, e.g., redacted text.

No.

### F. Are relationships between individual instances made explicit (e.g., users' movie ratings, social network links)? If so, please describe how these relationships are made explicit.

Yes. Every other sample contains the same prompt as the previous sample (e.g. sample 1 and 2, sample 3 and 4 etc).

### G. Does the dataset contain all possible instances or is it a sample (not necessarily random) of instances from a larger set? If the dataset is a sample, then what is the larger set? Is the sample representative of the larger set (e.g., geographic coverage)? If so, please describe how this representativeness was validated/verified. If it is not representative of the larger set, please describe why not (e.g., to cover a more diverse range of instances, because instances were withheld or unavailable).

All instances.

### H. Are there recommended data splits (e.g., training, development/validation, testing)? If so, please provide a description of these splits, explaining the rationale behind them.

20324 train and 1038 validation.
21362 total.
split based on 95% train and 5% val

### I. Are there any errors, sources of noise, or redundancies in the dataset? If so, please provide a description.

Yes, the annotated labels are from humans and the IRA (quadratic cohen's $\kappa$ is as follows)
helpfulness - 0.791 correctness - 0.793 coherence - 0.428 complexity - 0.427 verbosity - 0.548

*J. Is the dataset self-contained, or does it link to or otherwise rely on external resources (e.g., websites, tweets, other datasets)? If it links to or relies on external resources, a) are there guarantees that they will exist, and remain constant, over time; b) are there official archival versions of the complete dataset (i.e., including the external resources as they existed at the time the dataset was created); c) are there any restrictions (e.g., licenses, fees) associated with any of the external resources that might apply to a future user? Please provide descriptions of all external resources and any restrictions associated with them, as well as links or other access points, as appropriate.*

Yes self contained.

No.

## III. COLLECTION PROCESS

*A. What mechanisms or procedures were used to collect the data (e.g., hardware apparatus or sensor, manual human curation, software program, software API)? How were these mechanisms or procedures validated?*

Collected through Scale AI annotation platform.

We obtained prompts from ShareGPT and prompted internal LLM models to generate the response. The labels for the five attributes are from Scale AI annotators.

*C. If the dataset is a sample from a larger set, what was the sampling strategy (e.g., deterministic, probabilistic with specific sampling probabilities)?*

We sampled prompts by first clustering 90k shareGPT prompts into 1000 topics and uniformly sampled across each topic.

*D. Who was involved in the data collection process (e.g., students, crowdworkers, contractors) and how were they compensated (e.g., how much were crowdworkers paid)?*

Annotators from our vendor Scale AI - their compensation was handled by Scale AI but they guaranteed that it means local pay standards.

*E. Over what timeframe was the data collected? Does this timeframe match the creation timeframe of the data associated with the instances (e.g., recent crawl of old news articles)? If not, please describe the timeframe in which the data associated with the instances was created.*

Feb - Mar 2024. Yes - the responses to prompts were created in that period as well.

## IV. DATA PREPROCESSING

*A. Was any preprocessing/cleaning/labeling of the data done (e.g., discretization or bucketing, tokenization, part-of-speech tagging, SIFT feature extraction, removal of instances, processing of missing values)? If so, please provide a description. If not, you may skip the remainder of the questions in this section.*

We removed around 10% samples with contained high disagreement (2 points or larger on likert 5 scale) between annotators on its helpfulness value.

*B. Was the "raw" data saved in addition to the preprocessed/cleaned/labeled data (e.g., to support unanticipated future uses)? If so, please provide a link or other access point to the "raw" data.*

No.

No.

Yes.

No.

## V. DATASET DISTRIBUTION

HuggingFace Datasets.

14 June 2024. CC-BY-4.0.

No (it is CC-BY-4.0 Licensed)

No.

No.

## VI. Dataset Maintenance

### A. Who is supporting/hosting/maintaining the dataset?

HuggingFace Datasets.

### B. Will the dataset be updated? If so, how often and by whom?

No.

### C. How will updates be communicated? (e.g., mailing list, GitHub)

N.A.

### D. If the dataset becomes obsolete how will this be communicated?

We will update the README on Huggingface datasets.

### E. Is there a repository to link to any/all papers/systems that use this dataset?

No.

### F. If others want to extend/augment/build on this dataset, is there a mechanism for them to do so? If so, is there a process for tracking/assessing the quality of those contributions. What is the process for communicating/distributing these contributions to users?

No.

## VII. Legal and Ethical Considerations

### A. Were any ethical review processes conducted (e.g., by an institutional review board)? If so, please provide a description of these review processes, including the outcomes, as well as a link or other access point to any supporting documentation.

Yes. Ethical review was conducted by our vendor Scale AI as well as NVIDIA data collection teams.

### B. Does the dataset contain data that might be considered confidential (e.g., data that is protected by legal privilege or by doctorpatient confidentiality, data that includes the content of individuals non-public communications)? If so, please provide a description.

No.

### C. Does the dataset contain data that, if viewed directly, might be offensive, insulting, threatening, or might otherwise cause anxiety? If so, please describe why

No.

### D. Does the dataset relate to people? If not, you may skip the remaining questions in this section.

No.

### E. Does the dataset identify any subpopulations (e.g., by age, gender)? If so, please describe how these subpopulations are identified and provide a description of their respective distributions within the dataset.

### F. Is it possible to identify individuals (i.e., one or more natural persons), either directly or indirectly (i.e., in combination with other data) from the dataset? If so, please describe how.

### G. Does the dataset contain data that might be considered sensitive in any way (e.g., data that reveals racial or ethnic origins, sexual orientations, religious beliefs, political opinions or union memberships, or locations; financial or health data; biometric or genetic data; forms of government identification, such as social security numbers; criminal history)? If so, please provide a description.

### H. Did you collect the data from the individuals in question directly, or obtain it via third parties or other sources (e.g., websites)?

### I. Were the individuals in question notified about the data collection? If so, please describe (or show with screenshots or other information) how notice was provided, and provide a link or other access point to, or otherwise reproduce, the exact language of the notification itself.

### J. Did the individuals in question consent to the collection and use of their data? If so, please describe (or show with screenshots or other information) how consent was requested and provided, and provide a link or other access point to, or otherwise reproduce, the exact language to which the individuals consented.

### K. If consent was obtained, were the consenting individuals provided with a mechanism to revoke their consent in the future or for certain uses? If so, please provide a description, as well as a link or other access point to the mechanism (if appropriate).

### L. Has an analysis of the potential impact of the dataset and its use on data subjects (e.g., a data protection impact analysis)been conducted? If so, please provide a description of this analysis, including the outcomes, as well as a link or other access point to any supporting documentation.

### M. Any other comments?

## References