# OpenReview forum: "HelpSteer 2: Open-source dataset for training top-performing reward models"
_NeurIPS.cc/2024/Datasets_and_Benchmarks_Track — NeurIPS 2024 Track Datasets and Benchmarks Poster_

### Official Review · Reviewer_MsiT · 2024-07-15
**A high-quality open-sourced dataset for reward model training**

**Rating:** 6
**Confidence:** 3
**Clarity:** Yes in general, except for sth to cla…

**Review:**

This paper presents HelpSteer2, a permissively-licensed (CC-BY-4.0), small (10k pairs) and high quality dataset. Built upon this dataset, the paper is able to train reward models (5-head regression model) that perform very well on RewardBench. The paper further shows that the reward model can be used to align Llama 3 70B at least as well as Llama 3 70B Instruct and GPT-4-0613 on major alignment metrics.

Strength
- There are several careful choices made along the way that the reviewer feels are good choices. In detail, the prompts are sourced from ShareGPT + some proprietary prompts. Then data is sampled based on clustering and prompt difficulties. Then two responses are generated for each prompt. Multiple annotators are used for each data point to label five attributes in a pointwise manner.

- The performance on the popular rewardbench dataset looks strong. It is especially interesting to see that such performance can be achieved by pointwise annotations and the authors have some discussions around that.

- The alignment study looks good that it studied several alignment methods and evaluated on several tasks. These are good efforts to further verify the value of the trained reward models.


Weakness
- The reviewer is not sure about some filtering choices, such as the dataset being English only, and without coding tasks.

- The reviewer has mixed feeling about RewardBench. Is it possible to provide some statistical significance analysis by comparing with some methods that the authors have access to? It is possible that the improvements are not significant (which will not likely alter the reviewer's overall rating, but the reviewer is interested to understand the significance of the improvement).

- Given five predictions from the RM, the weights to combine them on rewardbench are tuned on rewardbench. This looks important enough to be *not* put in the appendix. To be fair, to the reviewer's knowledge, this is not uncommon, but still not ideal. Can the authors at least provide more discussions on the sensitivity of the weights? For example, what would the numbers look like when the grid search is at different granularity (currently it seems to be at 0.01, what about 0.1 and 1?)

- The reviewer did not fully get section 4.3, 4.4, 4.5. DPO uses helpsteer2 with human labels, then use Daring Anteater SFT with the reward model. PPO uses helpsteer2 with the reward model. SteerLM uses Daring Anteater SFT with the reward model. These look strange without further explanations (e.g., what if DPO just uses helpsteer2 with the reward model, similar to PPO, or the other way, where PPO uses similar data as DPO). These choices make Table4 harder to interpret. Can the authors provide some justifications of the choices?

I would be happy to raise the score if the questions can be clarified a bit.

**Strengths:**

- There are several careful choices made along the way that the reviewer feels are good choices. In detail, the prompts are sourced from ShareGPT + some proprietary prompts. Then data is sampled based on clustering and prompt difficulties. Then two responses are generated for each prompt. Multiple annotators are used for each data point to label five attributes in a pointwise manner.

- The performance on the popular rewardbench dataset looks strong. It is especially interesting to see that such performance can be achieved by pointwise annotations and the authors have some discussions around that.

- The alignment study looks good that it studied several alignment methods and evaluated on several tasks. These are good efforts to further verify the value of the trained reward models.

**Additional Feedback:**

na

**Correctness:**

The reviewer is overall positive about the choices made to construct the dataset.

**Documentation:**

Looks good.

**Ethics:**

The reviewer feels it's ok as the authors seem quite careful during the dataset construction.

**Limitations:**

English only and potential lack of diversity.

**Opportunities For Improvement:**

See detailed review.

**Relation To Prior Work:**

Looks good.

**Summary And Contributions:**

This paper presents HelpSteer2, a permissively-licensed (CC-BY-4.0), small (10k pairs) and high quality dataset. Built upon this dataset, the paper is able to train reward models that perform very well on RewardBench. The paper further shows that the reward model can be used to align Llama 3 70B at least as well as Llama 3 70B Instruct and GPT-4-0613 on major alignment metrics.

---

> ### Author Rebuttal · Authors · 2024-08-16
>
> Thank you for appreciating our high-quality dataset and your insightful comments!
>
> > The reviewer is not sure about some filtering choices, such as the dataset being English only, and without coding tasks.
>
> The generalist annotator pool (from our vendor) available to construct this dataset did not have capabilities to annotate responses to such prompts adequately. For instance, many coding prompts require expertise, equivalent to a few years of software engineering work experience. If such prompts are naively given to the annotator pool without the necessary expertise, annotators might not fully understand the response to give a reliable judgment of how good it is. In order to annotate such prompts, specialized annotator pools with a different set of skills would be required, which was not available to us at the start of this project. As a result, we skipped such prompts to guarantee the quality of annotations. We plan to address such a limitation in subsequent research.
>
> > The reviewer has mixed feeling about RewardBench. Is it possible to provide some statistical significance analysis by comparing with some methods that the authors have access to? It is possible that the improvements are not significant (which will not likely alter the reviewer's overall rating, but the reviewer is interested to understand the significance of the improvement).
>
> Providing statistical significance for Reward Bench is challenging since the benchmark does not output confidence intervals. Computing bootstrap-based confidence intervals should be possible in theory, but challenging/non-standard since the Reward Bench score is not just a basic average over all its tasks (there are multiple subsets with different weights), using the standard error over those tasks to derive a meaningful confidence interval would not be accurate. If you’re interested (in this inaccurate estimate), that would be sqrt(0.882(1-0.882)) / sqrt(2985) ~ 0.0059 or ~0.59%.
>
> To give you a further sense of expected variation, slight differences in LR can lead to ~0.5% difference in RewardBench Overall. This is minimal compared to the gap between the same base model (Llama 3 70B) trained on HelpSteer (66.1%) and Helpsteer2 (88.2%).
>
> This is based on our best effort interpretation of your request “comparing with some methods that the authors have access to?”, and if we have misunderstood this request, please let us know.
>
> > Given five predictions from the RM, the weights to combine them on rewardbench are tuned on rewardbench. This looks important enough to be not put in the appendix. To be fair, to the reviewer's knowledge, this is not uncommon, but still not ideal. Can the authors at least provide more discussions on the sensitivity of the weights? For example, what would the numbers look like when the grid search is at different granularity (currently it seems to be at 0.01, what about 0.1 and 1?)
>
> We acknowledge the importance of weights and will move them to the main paper in the revised version. We conducted sensitivity analysis on the Nemotron 4 340B models. If we compute the reward score based on a single attribute, Helpfulness gives 91.0% Overall RewardBench, Correctness 90.1% and Coherence 89.1%. Combining these attributes in a grid search gives 91.4% at granularity of 1, 91.5% at granularity of 0.1 and 91.6% at granularity of 0.01. Therefore, the RM is not very sensitive to weight search.
>
> > The reviewer did not fully get section 4.3, 4.4, 4.5. DPO uses helpsteer2 with human labels, then use Daring Anteater SFT with the reward model. PPO uses helpsteer2 with the reward model. SteerLM uses Daring Anteater SFT with the reward model. These look strange without further explanations (e.g., what if DPO just uses helpsteer2 with the reward model, similar to PPO, or the other way, where PPO uses similar data as DPO). These choices make Table4 harder to interpret. Can the authors provide some justifications of the choices?
>
> Our goal with Table 4/Sec 4.3-4.5 is to show how other researchers can leverage the strengths of HelpSteer2 and/or a Reward Model trained with HelpSteer2 to train LLMs using various well-known alignment techniques. Although we try to keep the data used for the three algorithms comparable, please note that:
> this is not always feasible as different algorithms require different types of training data, and
> we are *not* trying here to perform an exhaustive comparison between those algorithms to figure out which one works best (which we believe would require a more carefully controlled setup in terms of both dataset choice and hyperparameter tuning)
>
> If you’re specifically looking for an apples-to-apples comparison between DPO and PPO, both our vanilla DPO and PPO only use the HelpSteer2 dataset during the preference tuning stage. However, vanilla DPO does not use a reward model [1]: in order to leverage our reward model for DPO, we chose to follow the recipe from [2] to iteratively augment a preference set based on “pseudo-on-policy” sampling and scoring by a reward model, and it seemed natural to do so on a new set of prompts (like Daring Anteater) so as to broaden the prompt distribution. Although it would also be possible to train PPO on a combination of prompts from Helpsteer2 and Daring Anteater, we initially considered it out of scope for this limited study.
>
> > I would be happy to raise the score if the questions can be clarified a bit.
>
> We hope we have provided satisfactory answers to your questions and if we have not, please feel free to follow up and we will try our best to address them.
>
> [1] https://arxiv.org/abs/2305.18290
>
> [2] https://arxiv.org/abs/2405.07863

---

> ### Author Response · Authors · 2024-08-31
> **Follow up on Rebuttal**
>
> Hi Reviewer MsIT,
>
> With the end of the discussion period approaching today (31 Aug), we will appreciate if you can have a look at our rebuttal and let us know if we have clarified your initial questions satisfactorily. Otherwise, we are also happy to address any follow-up questions.

---

### Official Review · Reviewer_WxYS · 2024-07-23

**Rating:** 5
**Confidence:** 4
**Correctness:** Correct
**Clarity:** Yes

**Review:**

### Advantages

1. The authors implemented a robust annotation process involving multiple annotators (averaging 3.41 per sample), which enhances the reliability of the ratings.
2. HelpSteer2 achieves competitive performance with only 10,681 prompt-response pairs, making it highly efficient compared to larger datasets. This efficiency allows faster training and experimentation while maintaining high-quality outputs, which is crucial for researchers with limited resources.
3. By providing a CC-BY-4.0 licensed dataset, the authors encourage community engagement and further research. This openness not only promotes transparency in model training but also facilitates the development of aligned AI systems across both academic and commercial settings.

### Disadvantages
2. The filtering process for prompts based on language and coding expertise may inadvertently exclude valuable input types or contexts that could enhance the dataset's robustness. This selective approach might lead to a less comprehensive representation of real-world usage scenarios.
3. Although the multi-annotator strategy improves quality, it requires substantial resources and time. This meticulous process may not be feasible for all research teams, potentially limiting the dataset's adoption in broader contexts where rapid data collection is necessary.
4. Lacking of Analysis: More in-depth analysis on HelpSteer is strongly recommended. For example, the prompt, response distribution, and annotator pipeline are worth investigation.

**Strengths:**

See review

**Additional Feedback:**

None

**Documentation:**

Yes

**Ethics:**

No ethical concerns

**Limitations:**

See review.

**Opportunities For Improvement:**

See review

**Relation To Prior Work:**

Yes

**Summary And Contributions:**

The paper introduces HelpSteer2, a permissively licensed preference dataset (CC-BY-4.0) aimed at enhancing the training of reward models for large language models (LLMs). The authors emphasize the importance of high-quality preference datasets in aligning LLM outputs with human preferences.

---

> ### Author Rebuttal · Authors · 2024-08-16
>
> Thank you for feedback on our paper!
>
> > 2. The filtering process for prompts based on language and coding expertise may inadvertently exclude valuable input types or contexts that could enhance the dataset's robustness. This selective approach might lead to a less comprehensive representation of real-world usage scenarios.
>
> We agree that excluding prompts that are non-English or require coding expertise can reduce the dataset's robustness to such inputs. The generalist annotator pool (from our vendor) available to construct this dataset did not have capabilities to annotate responses to such prompts adequately. For instance, many coding prompts require expertise, equivalent to a few years of software engineering work experience. If such prompts are naively given to the annotator pool without the necessary expertise, annotators might not fully understand the response to give a reliable judgment of how good it is. In order to annotate such prompts, specialized annotator pools with a different set of skills would be required, which was not available to us at the start of this project. As a result, we skipped such prompts to guarantee the quality of annotations. We plan to address such a limitation in subsequent research.
>
> > 3. Although the multi-annotator strategy improves quality, it requires substantial resources and time. This meticulous process may not be feasible for all research teams, potentially limiting the dataset's adoption in broader contexts where rapid data collection is necessary.
>
> We acknowledge that a resource-intensive, multi-annotator, meticulous strategy focussing on quality may not be feasible for all research teams. This is the reason we release HelpSteer2 with a fully permissive CC-BY-4.0 license, contributing to the research community, and avoiding the need for the community to repeat such a resource-intensive data collection process.
> Therefore, we see our "meticulous process" as a strength of our contribution rather than a weakness, and we would like to ask the Reviewer what they meant by "limiting the dataset's adoption in broader contexts where rapid data collection is necessary" since our dataset can be readily adopted without any additional data collection.
>
> > 4. Lacking of Analysis: More in-depth analysis on HelpSteer is strongly recommended. For example, the prompt, response distribution, and annotator pipeline are worth investigation.
>
> **Prompt Distribution:**
>
> The distribution of prompts across each area follows the distribution of topics within the ShareGPT data (proxy of real world usage of ChatGPT). We performed a hierarchical topic model of the HelpSteer2 data, which gives the following top 7 overarching topics (we will add this analysis to the paper):
>
> 1. General/Scientific 33.2%
> 2. Long form text generation (Dialogue/Document/Poem) 26.5%
> 3. Coding-related (but doesn’t require code output) 10.7%
> 4. Marketing/HR 10.3%
> 5. Technology 8.4%
> 6. Finance/Business 5.6%
> 7. Education 3.8%
>
> **Annotator Pipeline:**
>
> Compared to platforms such as Amazon Mechanical Turks or Prolific that mostly serve as annotator recruitment platforms, our vendor Scale.AI provides a sophisticated  proprietary annotation pipeline to detect and address issues at various stages of annotation. Unfortunately, we are not able to share more details of the proprietary annotator pipeline of our vendor Scale.AI since it constitutes sensitive business information.
>
> **Response Distribution:**
>
> We have included analyses of responses in Table 2 and Lines 74-97. Please let us know if there are further specific analyses that you would like.
>
>
> > Overall
>
> We hope we have addressed most of your concerns and would appreciate if you could reconsider your assessment of the paper in light of this additional information.

---

> > ### Comment · Reviewer_WxYS · 2024-08-31
> >
> > Thanks for your rebuttal, I will raise my score

---

> > > ### Author Response · Authors · 2024-08-31
> > > **Thank you**
> > >
> > > Thank you for raising our score in light of our rebuttal! We noticed that the rating indicated on the review remains at "5: Marginally below acceptance threshold" before and after this comment, which communicates an increase. In case it slipped through due to a technical glitch, please take note to update the rating by end of day today(i.e. 31 Aug, which is the end of the discussion period) to avoid confusion for others.

---

### Official Review · Reviewer_BUY9 · 2024-07-24
**Good Results for RLHF**

**Rating:** 7
**Confidence:** 4
**Clarity:** This paper is well writen and easy to…

**Review:**

This paper presents a valuable preference dataset. Reward models trained with this preference dataset achieve state-of-the-art performance, while alignment methods using this preference dataset achieve promising results. Overall, this paper is of high quality.

**Strengths:**

The released preference dataset is helpful for researchers in this community. Furthermore, extensive alignment experiments have been conducted to validate the value of this dataset.

**Additional Feedback:**

None.

**Correctness:**

The claims (Lines 210-226) about the advantages over Bradley-Terry (BT)-based reward models are not well supported. More experiments and ablation studies are needed to validate these claims.

**Documentation:**

Yes.

**Ethics:**

None.

**Limitations:**

Yes

**Opportunities For Improvement:**

From Table 4, it seems that the SFT dataset, Daring Anteater, is very powerful. Specifically, the Llama-3-70B model trained via SFT with this dataset can achieve similar performance to the official Llama-3-70B-Instruct model. Actually, the further improvements by alignment methods such as DPO and PPO are not too significant. It is expected to explore and evaluate the improvements on downstream tasks such as code generation, math reasoning tasks, and tool usage to see the significance of the preference dataset.

**Relation To Prior Work:**

This paper provides sufficient discussion of prior work.

**Summary And Contributions:**

This paper presents a new preference dataset called HelpSteer2. The reward model trained with this preference dataset achieves state-of-the-art performance on the RewardBench benchmark. Furthermore, this paper also implements iterative DPO, PPO, and Steer LM to verify the effectiveness of the preference dataset and reward model.

---

> ### Author Rebuttal · Authors · 2024-08-16
>
> Thank you for your recognition of this project and the helpful comments!
>
> > From Table 4, it seems that the SFT dataset, Daring Anteater, is very powerful. Specifically, the Llama-3-70B model trained via SFT with this dataset can achieve similar performance to the official Llama-3-70B-Instruct model. Actually, the further improvements by alignment methods such as DPO and PPO are not too significant.
>
> We agree that the Daring Anteater dataset is very powerful since it is generated mostly by Nemotron-4 340B. Hence, it brings in a strong distillation effect. In our experiments as well as findings from [1], preference learning provides limited improvement when strong distillation happens during SFT. In contrast, when SFT data is weaker (e.g. Open Assistant), we do observe significant improvement with advanced alignment algorithms incorporating preference information (e.g. SFT w/ Open Assistant scores 6.75 on MT bench, while SteerLM scores 7.44).
>
> > It is expected to explore and evaluate the improvements on downstream tasks such as code generation, math reasoning tasks, and tool usage to see the significance of the preference dataset.
>
> Regarding code generation, reasoning, and tool use, as these areas are outside the scope of the HelpSteer2 dataset and no related prompts are present in the data, we do not anticipate gains in these areas.
>
> > The claims (Lines 210-226) about the advantages over Bradley-Terry (BT)-based reward models are not well supported. More experiments and ablation studies are needed to validate these claims.
>
> We agree that “the claims (Lines 210-226) about the advantages over BT reward models are not well supported”. We would like to provide additional results backing up these claims. We trained a BT RM on Helpsteer2 from the Nemotron-4 340B base model. We experimented with various learning rates and batch sizes, and found it best – in terms of validation accuracy – to increase the batch size from 128 to 256, along with the learning rate to from 7e-7 to 5e-6 and decrease the number of epochs from 2 to 1 (since it overfits otherwise, see [2] for insights about overfitting issues with BT reward models). Our main observations are as follows:
>
> 1. We observe that the BT model chooses the longer response as being better (on the HS2 validation split) 63.6% of the time, vs. 50.7% for SteerLM RM, which supports our claim that “binary-trained reward models might sometimes incorrectly associate ‘goodness’ with artifacts like response length”, while the SteerLM model tends to “disambiguate verbosity from the overall quality of the response”.
>
> 2. On Reward Bench, this BT model scores 82.2 overall (chat: 98.3, chat_hard: 63.3, safety: 73.9, reasoning: 94.3), vs. 91.6 for our SteerLM model (chat: 95.5, chat_hard: 86.4, safety: 90.8, reasoning: 93.6). Interestingly, on the first three categories, where both models’ scores differ significantly, we observe that:
>      - The BT model wins on the category biased towards preferring longer responses (“chat”, where 79% of the Reward Bench preferences favor the longer response)...
>      -  …but under-performs on the categories where shorter responses are often preferred (“chat_hard” and “safety”, where respectively 71% and 58% of the Reward Bench preferences favor the shorter response)
>
> 3. Those observations are in line with the previous point, and suggest that this BT model is relying (at least to some extent) on spurious length correlations, which makes it unsuitable for some tasks like those from “chat_hard” (which arguably is more interesting than the easier “chat” category, as it is designed to require a more careful analysis of the responses being compared in order to identify the best one). On the other hand, our SteerLM model, by disentangling the various attributes of the response – including verbosity – appears more robust to such spurious correlations.
>
> 4. To support our argument in l.219-226 regarding the prompt-dependent reward offset of the BT model, we compare the accuracy on the HelpSteer2 validation set of both models on two tasks:
>      - Given two responses to the same prompt, identify (by the reward score) which response has highest overall helpfulness rating (according to ground truth human annotations): this is the “standard” validation metric, and the BT model scores 76%, while the SteerLM model scores 74.3% (this difference is within noise level, but seeing the BT model perform slightly better is not surprising considering that the Helpsteer 2 validation set also has a moderate bias towards longer responses, with the highest helpfulness response being the longest 61% of the time)
>      - Do the same but given two responses to *different* prompts: the BT model’s score drops to 69.1%, while the SteerLM score remains stable at 74.4%. This suggests that SteerLM-based rewards are more comparable across prompts than the BT rewards.
>
> We acknowledge that these findings may not necessarily generalize to other datasets, or to algorithmic variants of BT training: a more thorough empirical comparison of BT vs. SteerLM reward models would be required to draw more definitive conclusions, as well as evaluations on downstream tasks of models aligned with those reward models. We will clarify such claims as preliminary hypotheses.
>
> [1] https://arxiv.org/pdf/2402.12366
>
> [2] https://arxiv.org/pdf/2401.16335

---

> > ### Comment · Reviewer_BUY9 · 2024-09-02
> >
> > I appreciate the authors' clarification. I decide to maintain my acceptance recommendation.

---

### Official Review · Reviewer_67qs · 2024-07-25
**a noteworthy contribution to the research community**

**Rating:** 8
**Confidence:** 4
**Clarity:** The paper is well written.

**Review:**

**Pros:**
- Open-sourcing a dataset with a sizable amount of human annotations (~21.3k) is always welcomed by the community, thanks for your contributions.
- Very transparent about the details of prompts & annotation collections, and data filtering process. Documentations (including annotation methods and annotation guidelines) are provided in detail.
- The procedure for collecting the model responses is sound. Using multiple sources for response generation to ensure diversity (86.2% - Own Internal LLMs, parameter size by billions: [8, 15, 22, 43, 340]. Nemotron-2 = 18.9%, Nemotron-3 = 40.4%, Nemotron-4 = 26.9%, 7.9% - Mixtral-8x7B-Instruct, 5.9% Human annotators from Scale AI). This is much more diverse than the original HelpSteer dataset, which only generates from Nemotron-2 43B.
- The procedure for collecting human feedback is sound. Contracted ~1000 human annotators from Scale AI, and assigned at least three annotators to each response. Tie-breaking process is also thoroughly described - if display a high level of disagreement (i.e. score difference among them is > 2), two additional annotators are recruited. On average, samples were annotated by 3.41 annotators.
- The analysis for inter-annotator agreement (Table 1) and descriptive statistics for attributes (Table 2) looks correct and insightful. The improvements from HS1 to HS2 shown in Table 2 look very significant.
- Able to train a SOTA reward model on RewardBench on top of the Llama-3 base model is remarkable. The authors are also being transparent about the low prior set performance and have provided a convincing argument for it.

**Cons:**
- The improved SFT method, SteerLM2, seems unnecessarily complicated in derivation (e.g. approximating probability with beta distribution, how rigorous can this be?), and looking at Table 4, SteerLM 2 does not bring significant, if not backward, performance improvements. Arguably, most improvements are driven by the introduction of the new SFT dataset (DA) and the new RM dataset (HS2).
- If possible, I strongly encourage the authors to release a hf-transformers version of your Nemotron 4 340B models. It's not only good for spreading the influence of your work, but it removes roadblocks for the community to adopt your models.

Overall I think this is a very meaningful contribution to our community. Therefore I am very pleased to vote for the acceptance of this paper.

**Strengths:**

See Review.

**Additional Feedback:**

N/A

**Correctness:**

The dataset is constructed in a sound way. I binarized the dataset and trained a Bradley-Terry reward model out of it, and was able to attain good accuracy on the test set. Though I have not attempted to reproduce the experiment result in the paper.

**Documentation:**

Documentation is sufficient.

**Ethics:**

Not particularly worried. It should be noted that the authors have asked the annotators will skip all samples containing Personally Identifiable Information and flag all unsafe content. I have not investigated how effective this is.

**Limitations:**

Limitations, societal impact, and ethical considerations have been adequately discussed in the first three sections of the appendix. I agree with most of the points the authors raised in the limitation section.

**Opportunities For Improvement:**

- Line 280 indicated that the authors handpicked a particular convex combination of the weightings of the five objectives in the PPO reward. This is acceptable as this can also be a hyperparameter to tune, but one can also argue that this is simply hacking the RewardBench. It would be interesting if we could see a sensitivity study between all the weights and the evals. For example, why did you completely discard verbosity and complexity? What interactive effects do the KL penalty and the verbosity objective have on each other?
- Very minor, but the lack of intuitive graphical figures might hurt the readability and the overall aesthetics of this paper.

**Relation To Prior Work:**

Most related work has been appropriately cited. A small suggestion might be to add background knowledge about SFT, RLHF, BT v.s. attributed-based reward models.

**Summary And Contributions:**

HelpSteer 2 highlights three contributions:
1. 340B reward model with an NVIDIA Open Model License, which is *"a permissive license that allows distribution, modification, and use of the Nemotron-4 340B models and its outputs for personal, research, and commercial use, without attribution requirements"*
2. a CC-BY-4.0-licensed open-source chat dataset, with a mix of single- and (~29% of the dataset) multi-round conversations and rating the response on five key attributes (helpfulness, correctness, coherence, complexity, verbosity) on a Likert-5 scale. Prompts are >95% response-stripped SharedGPT and a small portion of proprietary prompts.
3. An improved SFT algorithm extended from SteerLM (their previous work)


The resulting reward model, which is Nemotron-4 340B RM, is models as a multi-objective reward model, which is a base model + a linear layer that converts the final layer representation of the EOS token into five scalar float values ranging from 0 to 4 (each corresponding to a HelpSteer2 attribute). It was fine-tuned on top of the Nemotron-4 340B base model and trained 2 epochs on HS2 using the regular MSE loss function.

---

> ### Author Rebuttal · Authors · 2024-08-16
>
> Thank you for your appreciation of our paper and thoughtful suggestions!
>
> > The improved SFT method, SteerLM2, seems unnecessarily complicated in derivation (e.g. approximating probability with beta distribution, how rigorous can this be?), and looking at Table 4, SteerLM 2 does not bring significant, if not backward, performance improvements. Arguably, most improvements are driven by the introduction of the new SFT dataset (DA) and the new RM dataset (HS2).
>
> We agree that the derivation of SteerLM2 may appear complex.   We chose to include the full derivation in the appendix to maintain focus on key insights and results in the main text. We would like to clarify that the beta distribution approximation is not an integral part of the SteerLM method itself, but rather a tool to convert our regression-based reward model into a probabilistic one. While this approximation has limitations, it was necessary.
>
> Yes, significant improvements come from new datasets. However, SteerLM2 adds meaningful contributions, achieving the best MT-Bench among all methods tested (despite modest gains on other metrics). We view the combination of improved datasets and SteerLM2 as a strength, creating a robust alignment strategy. We will make the presentation more accessible and emphasize specific contributions in the main text, while highlighting the need for further research into true probabilistic reward models.
>
> > If possible, I strongly encourage the authors to release a hf-transformers version of your Nemotron 4 340B models. It's not only good for spreading the influence of your work, but it removes roadblocks for the community to adopt your models.
>
> We have Nemotron 4 340B HF-compatible checkpoint in the works. It took us (and our partners from HF) some time because HF only recently supported multi-node inference, which this model needs. We'd also like to mention that anyone can use this model (without needing to login) on build.nvidia.com and a free API is also available.
>
>
> > Line 280 indicated that the authors handpicked a particular convex combination of the weightings of the five objectives in the PPO reward. This is acceptable as this can also be a hyperparameter to tune, but one can also argue that this is simply hacking the RewardBench. It would be interesting if we could see a sensitivity study between all the weights and the evals. For example, why did you completely discard verbosity and complexity? What interactive effects do the KL penalty and the verbosity objective have on each other?
>
> We did some sensitivity analysis on the Nemotron 4 340B Reward model. If we compute the reward score based on a single attribute, Helpfulness gives 91.0% Overall RewardBench, Correctness 90.1% and Coherence 89.1%, compared to the final combination at 91.6%.  Given single attribute performance being so close to the final combination, we think that the likelihood of hacking Reward Bench is low.
>
> We also agree that a sensitivity study between KL penalty and verbosity (or other attributes) is interesting but investigating it was outside of our scope as a dataset paper. Additionally, we attempt to not bias our policy by setting a non-zero verbosity weight. Our internal human evaluations have found that some cases having shorter response lengths are better (for example answering multiple choice questions) whereas in other cases having longer responses are preferred (for open-ended questions requiring more details).
>
> > Very minor, but the lack of intuitive graphical figures might hurt the readability and the overall aesthetics of this paper.
>
> We will take note and attempt to add some visuals into our paper.
>
>
> > A small suggestion might be to add background knowledge about SFT, RLHF, BT v.s. attributed-based reward models.
>
> Sure, we are happy to add further background information for these areas.

---

### Author Response · Authors · 2024-08-29
**End of discussion period approaching in 3 days**

With the end of the discussion period approaching in 3 days (31 Aug), we would appreciate if you can have a look at our rebuttal to see we have addressed your initial concerns. Otherwise, we are happy to provide additional details.

---

### Decision · Program_Chairs · 2024-09-26

**Decision:**

Accept (Poster)

**Comment:**

This paper presents HelpSteer 2, a new dataset for aligning large language models (LLMs) with human preferences. The dataset is constructed using a multi-annotator strategy and a variety of quality controls. The authors also introduce SteerLM 2, an improved supervised fine-tuning algorithm. The resulting dataset and algorithm are shown in experiments to lead to state-of-the-art models (per Reward Bench).

Reviewers appreciate the high quality of the dataset guaranteed by the use of multiple annotators. They noted the strong performance of the resulting model and the permissiveness of the licenses. Some reviewers raised concerns, for example, regarding the exclusion of prompts involving programming code or non-English language, but the authors generally explained well why certain trade-offs had to be made. Overall most reviewers agree that this dataset was well-constructed, results in models with impressive performance, and that the community will benefit from having this dataset and the resulting models publicly available. I recommend accepting this paper.